developmental biology, evolution

eye evolution, neofunctionalization, lens, cephalopod, Spiralia, transcription factor

**Author for correspondence:**
Kristen M. Koenig
e-mail: kmkoenig@fas.harvard.edu

# Krüppel-like factor/specificity protein evolution in the Spiralia and the implications for cephalopod visual system novelties

Kyle J. McCulloch[1,2] and Kristen M. Koenig[1,2]

[1]Department of Organismic and Evolutionary Biology, Harvard University Cambridge, MA 02138, USA
[2]John Harvard Distinguished Science Fellows, Harvard University, Cambridge, MA 02138, USA

KMK, 0000-0001-6093-2262

The cephalopod visual system is an exquisite example of convergence in biological complexity. However, we have little understanding of the genetic and molecular mechanisms underpinning its elaboration. The generation of new genetic material is considered a significant contributor to the evolution of biological novelty. We sought to understand if this mechanism may be contributing to cephalopod-specific visual system novelties. Specifically, we identified duplications in the Krüppel-like factor/specificity protein (KLF/SP) sub-family of C2H2 zinc-finger transcription factors in the squid *Doryteuthis pealeii*. We cloned and analysed gene expression of the KLF/SP family, including two paralogs of the *DpSP6-9* gene. These duplicates showed overlapping expression domains but one paralog showed unique expression in the developing squid lens, suggesting a neofunctionalization of *DpSP6-9a*. To better understand this neofunctionalization, we performed a thorough phylogenetic analysis of SP6-9 orthologues in the Spiralia. We find multiple duplications and losses of the *SP6-9* gene throughout spiralian lineages and at least one cephalopod-specific duplication. This work supports the hypothesis that gene duplication and neofunctionalization contribute to novel traits like the cephalopod image-forming eye and to the diversity found within Spiralia.

## 1. Introduction

The generation of new genetic material subject to mutation is considered a significant contributor to the evolution of biological novelty [1]. This can include large genomic expansions and rearrangements, gene duplications and *de novo* gene emergence, *cis*-regulatory element expansion or exon shuffling [2–4]. However, a lack of sequence data has made it difficult to assess the importance of this mechanism in most metazoan species until recently. Cephalopods, a group of molluscs that include squid, cuttlefish, octopus, and nautilus, have recently benefited from genome and transcriptome sequencing and are exceptional subjects for the study of morphological novelty because they have evolved a diversity of complex, class-specific traits [5–12].

Cephalopod novelties include many features traditionally associated with vertebrates (closed circulatory system, single-chambered eyes) as well as some defining to themselves (ink decoys and textured, camouflaging skin). Little is known about the molecular or developmental basis of these traits, including whether the genetic and genomic changes that underpin these novelties are similar to the mechanisms found in better-studied species. Cephalopod genomes are large, ranging between two and five gigabases [13]. Sequencing the octopus genome found evidence of large-scale transposon-mediated gene family expansions [10]. One of these expansions is the large C2H2 zinc-finger superfamily of proteins, with over double the number of homologues found

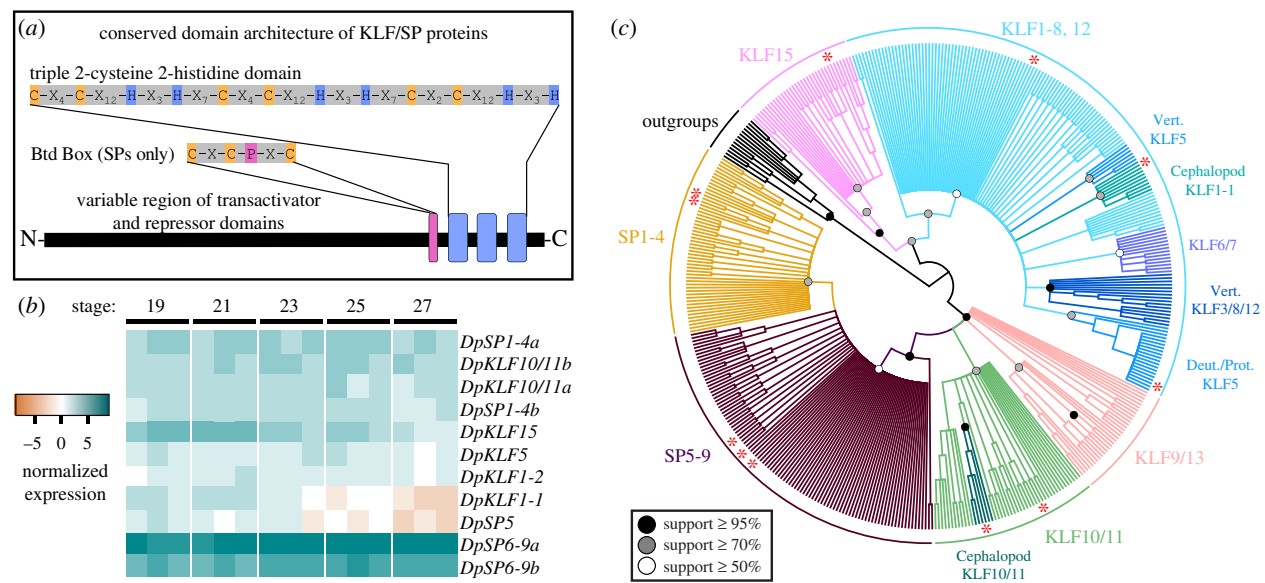

**Figure 1.** Domain architecture, expression, and phylogeny of KLF/SP family members (a) KLF/SP proteins are defined by a conserved triple 2-cysteine 2-histidine (C2H2) DNA-binding domain. The Buttonhead box (Btd) distinguishes SP transcription factors from KLFs. (b) Normalized expression of KLF/SP family in the developing eye and optic lobe in *D. pealeii*, stages 19, 21, 23, 25, and 27 in biological triplicate (original dataset from [6]). *DpSP6-9a* and *DpSP6-9b* are highly expressed throughout development. (c) Bayesian inference of KLF/SP phylogeny (MrBayes). Many deeper nodes are poorly supported, making it difficult to draw firm conclusions about the relationships of many KLF/SP subgroups. Circles represent posterior probabilities above 0.5 (white), 0.7 (grey), and 0.95 (black) on branches leading to the nodes of major labelled and coloured subclades. Subgroup naming conventions from previous phylogenetic studies have been used. Red asterisks mark *D. pealeii* sequences in the tree.

in vertebrates. This is consistent in the genome of the bobtail squid, *Euprymna scolopes*, and in the transcriptome of the longfin squid, *Doryteuthis pealeii* [6,8]. To understand the contribution of C2H2 zinc-finger protein family expansion to the evolution of cephalopod novelty, we have chosen to focus on a sub-family with well-understood function in vertebrates and *Drosophila*: the KLF (Krüppel-like factor) and SP (specificity protein) genes [14–17].

KLF/SP genes comprise a family of related transcription factors found within the Holozoa characterized by a conserved triple C2H2 zinc-finger DNA-binding domain at the C-terminus. The SP proteins are restricted to the Metazoa and are defined by a Buttonhead box (Btd), a putative transactivating domain, and usually an SP box located at the N-terminus (figure 1a; electronic supplementary material, figure S1) [18]. SP1 was the first eukaryotic transcription factor identified. Since the discovery of SP1, KLF/SP proteins have been implicated in diverse developmental, cellular, and homeostatic processes [19,20]. They have known roles as both transcriptional activators and repressors in neural development, angiogenesis, osteogenesis, muscle, digestive, and renal physiology, as well as in cancer progression [19–26]. The KLF/SP triple C2H2 domain binds to GC/GT boxes in DNA and its biochemical function is conserved across distantly related species [27]. Partial functional redundancy is often observed in overlapping paralog expression, suggesting that the context of expression is important for KLF/SP functional diversity rather than molecular differences [28]. There are 17 KLF and 9 SP genes in mice and 5 KLF and 3 SP genes in *Drosophila* with variable numbers identified in other vertebrates and ecdysozoans [29,30]. However, the phylogeny and function of this group of genes is poorly understood in the Spiralia. Analysis of previously published RNA-seq data in the squid *D. pealeii* [6] shows an enrichment of KLF/SP family members in the developing visual system, including the eye and optic lobe tissues, where

image processing occurs (figure 1b) [31]. The visual system is a compelling subject for the study of biological novelty because it demonstrates significant innovations in coleoid, or soft-bodied, cephalopods [32–34].

Cephalopods are one of four groups of animals, including vertebrates, pancrustaceans, and arachnids, that have independently evolved a highly acute visual system [35]. Coleoid cephalopod eye morphology is physically similar to the vertebrate eye, composed of a single chamber with a lens and a cup-shaped retina (electronic supplementary material, figure S2). This complex organ evolved within Cephalopoda and is convergent with other camera-type eyes. The cephalopod lens is a lineage-specific morphological novelty that allows for high spatial resolution by precisely focusing light on the retina [36–38]. The cephalopod eyes develop from two placodes on either side of the embryo early in development. These placodes are then internalized by a lip of cells to form the optic vesicles. The posterior of the optic vesicle will generate the retina and the anterior will form the lens and anterior segment of the eye. At hatching, the neural retina is composed of two morphologically identifiable cell types: photoreceptors, which synapse directly on the optic lobe, and support cells. The anterior segment of the eye is composed of multiple populations of lens-generating or lentigenic cells. The lens is segmented into an anterior lens segment and a posterior lens segment. The lentigenic cells are arranged circumferentially around the lens and selectively contribute to each segment (electronic supplementary material, figure S2) [6,39,40]. Each population of lentigenic cells extends long processes that wrap around each other to form the lens fibres [39,40]. Gene expression studies and staged RNA-seq data of the developing eye and optic lobe have been published but we have little understanding of the molecular contributors to lens development in cephalopods [6,34,39,41–47].

Our goal in this study is to illuminate if gene duplications in the KLF/SP transcription factor family may be associated

with visual system innovations in the cephalopod. Here, we identify duplications of the KLF/SP family of genes specific to cephalopods by generating a new phylogeny using recent genomic and transcriptomic data across Spiralia. To better understand the evolutionary history of one of these cephalopod duplications we focused on the SP6-9 homologues, which are enriched in the RNA-seq time course of the eye and optic lobe development. Our results show that the SP6-9 gene has undergone multiple duplications and losses within Spiralia including a cephalopod-specific duplication. We cloned and performed *in situ* hybridization on members of the KLF/SP family to assess their expression in the squid visual system. Our study showed lens-specific expression of a single paralog of the *DpSP6-9* gene identifying a potential neofunctionalization. This work supports the hypothesis that gene duplication plays an important role in the evolution of novelty and diversity found in the Spiralia.

## 2. Material and methods

### (a) Animal husbandry
Egg sacs were obtained from the Marine Biological Laboratory, Woods Hole, Massachusetts. Embryos were kept in flowing seawater at the MBL or at 20°C in artificial sea water.

### (b) Cloning and *in situ* hybridization
Genes were identified using a previously assembled and published transcriptome [6]. RNA was extracted from pooled stages of embryos using Trizol. cDNA libraries were generated using iScript according to the manufacturer's instructions (Bio-Rad). Primers used for cloning gene fragments are found in electronic supplementary material, table S1. PCR product size was confirmed by electrophoresis and then cloned into pGEM-T Easy and Sanger sequenced (Promega). Sense and antisense probes were generated from plasmid using digoxygenin-labelled ribonucleoside tri-phosphate (rNTPs) (Roche). *In situ* hybridization was performed as previously described [6]. Embryos were embedded and cryosectioned as previously described [6]. Whole-mount embryos were imaged using a Zeiss Axio Zoom and sectioned embryos were imaged using a Zeiss Axioskop 2.

### (c) Phylogenetic analysis
The metazoan KLF/SP family member trees were generated from amino acid sequences using previous alignments [29,30], and NCBI BLAST searches in additional spiralians and basal deuterostomes. We required that sequences had three C2H2 domains with the following 88 amino acid architecture: $C-X_4-C-X_{12}-H-X_3-H-X_7-C-X_4-C-X_{12}-H-X_3-H-X_7-C-X_2-C-X_{12}-H-X_3-H$, where X can be any amino acid. Additionally, we confirmed the presence of a conserved aspartic acid residue at position 44 ($D_{44}$) [30]. Non-metazoan, opisthokont KLF sequences were used as outgroups. Amino acid sequences were truncated and concatenated to include conserved activator/repressor motifs (SID, R2, R3, SP-box, Btd, and C2H2) if present. All sequences were aligned using MUSCLE and trees were visually inspected in Geneious [48]. We constructed maximum-likelihood (ML) and Bayesian trees run on the FASRC Cannon cluster supported by the FAS Division of Science Research Computing Group at Harvard University. To generate an ML tree, we used PTHREADS RAxML v. 8.2.10 run with default options and the PROTGAMMAAUTO model of amino acid substitution [49]. The best-scoring substitution model under the gamma model of rate heterogeneity was the LG model with fixed base frequencies. We resampled the tree with 1000 rapid bootstrap replicates and 500 best ML tree searches. For Bayesian analysis, we used MrBayes v. 3.2.6 [50] with default settings except the following: 10 million Markov chain Monte Carlo (MCMC) generations, with 1.5 million generations of burn-in, five heated chains per run, and 0.1 heating temperature. Stationarity was assessed by the convergence of the two runs using an average standard deviation of split frequencies which reached 0.042.

We identified SP6-9 sequences for our spiralian tree by reciprocal BLAST of available spiralian transcriptomes and genomes. We used full-length amino acid sequences for the SP6-9 trees when possible and used SP1-5 sequences as outgroups. We allowed the RAxML software to identify the best-scoring protein substitution model (PROTGAMMAAUTO) for our dataset and the JTT model with fixed base frequencies under the gamma model of rate heterogeneity was used. RAxML and MrBayes were otherwise implemented as above on the spiralian SP6-9 alignment. In the Bayesian analysis, the average standard deviation of split frequencies reached 0.096 after 10 million generations.

We built the cephalopod-only SP6-9 tree using the cephalopod amino acid sequences from our spiralian tree with vertebrate and *D. pealeii* SP1-5 sequences as outgroups. We aligned sequences in Muscle and implemented MrBayes as described above, for 565 000 generations until the average standard deviation of split frequencies was consistent at 0.019.

For the Mollusca and Annelida SP6-9 only trees, we trimmed, filtered, and assembled basally branching annelid and mollusc transcriptomes according to the Harvard FAS Informatics best practices, a publicly available pipeline found here: https://informatics.fas.harvard.edu/best-practices-for-de-novo-transcriptome-assembly-with-trinity.html. We used reciprocal BLAST to identify potential SP6-9 orthologues, aligned only sub-group specific sequences, and implemented MrBayes as above, with 4 chains in each of 2 runs, and 4 million generations. Alignment and nexus tree files may be found in supplementary data (electronic supplementary material, data S9–S12)

### (d) Analysis of SP6-9 synteny
Using publicly available spiralian genomes, we identified genomic coordinates and intron/exon boundaries of SP6-9 paralogs. We used all available cephalopod genomes. To assess synteny in other spiralians, we used chromosome-level assemblies with the exception of annelids which lack a chromosome-level assembly [51].

### (e) Character mapping duplications and losses of SP6-9
Cartoon cladograms were made by manually combining the most current hypotheses of relationships among Spiralia, Annelida, and Mollusca [52–61]. For simplicity, we only include branches for which we have data. We then mapped gains and losses to find the most parsimonious hypothesis. In the cases of Rotifera, Sedentaria, and Hypsogastropoda, we override maximum parsimony in favour of the phylogenetic signal we see from our gene trees.

## 3. Results and discussion

### (a) Analysis of *Doryteuthis pealeii* Krüppel-like factor/specificity protein homologs
We identified *DpKLF/SP* genes in a previously published *D. pealeii* transcriptome from reciprocal BLAST [6]. Time course developmental expression in the eye and optic lobe of these putative *DpKLF/SP* genes is shown in figure 1*b* [6]. Each DpKLF/SP amino acid sequence has the conserved triple C2H2 domain and the DpSP sequences have the requisite Btd box (electronic supplementary material, figure S1a). We also find previously identified transactivation and

repressor domains in these *D. pealeii* sequences, including the SID, PVDLS, R2, and R3 repressor domains and the 9aaTAD transactivating domain (electronic supplementary material, figure S1b) [30].

To validate the putative *DpKLF/SP* genes from our dataset, we constructed phylogenetic trees with representative basal metazoan and bilaterian KLF/SP family members (electronic supplementary material, table S2). Previous phylogenetic analyses either omitted or had limited sampling within the Spiralia [29,30]. This gene family presents a challenge for phylogenetic analysis due to the highly divergent N-terminal regions and the highly conserved but short C2H2 domains [17,30,62]. To produce the alignment, we concatenated any conserved activator and repressor motifs, the C2H2 domain, and Btd box. A summary of the Bayesian tree generated for this gene family is labelled and coloured by major subgroups supported by posterior probabilities (figure 1c). ML methods never generated a tree with support for all previously identified KLF/SP groups (electronic supplementary material, data S2). In our Baysian tree, we find support for sub-families: KLF15, KLF1-8+12, KLF9/13, KLF10/11, SP1-4, and SP5-9. Within Spiralia, we find most taxa have one KLF15 representative, four KLF1-8+12 members, a single KLF9/13 member, one SP1-4, and three SP5-9 members.

Our tree supports subgroup clades but the addition of understudied taxa does not resolve 1 : 1 orthology for many sequences. We found one *D. pealeii* KLF15 and three KLF1-8+ 12 family members. Within the KLF1-8+12 sequences, previously identified jawed vertebrate subgroups KLF5, KLF6/7, and KLF3/8/12 are well-supported, but the vertebrate KLF1/2/4 group was not resolved [15,63,64]. We cannot confidently assign orthology to many protostome KLF1-8+12 sequences. KLF6/7 is an exception and contains representatives from all metazoan lineages except Spiralia. KLF5 is not supported as a monophyletic group. We find a vertebrate KLF5 clade and an invertebrate KLF5 clade, which includes *D. pealeii*. Additionally, despite only finding support for a vertebrate KLF3/8/12 clade, which is defined by a conserved PVDLS domain, there are *Drosophila* and spiralian sequences with a PVDLS domain. Cephalopods have two KLF1-8+12 sequences and one KLF5. Aside from DpKLF5, the *D. pealeii* KLFs in this subgroup are too divergent to confidently assess orthology, so we name them DpKLF1-1 and DpKLF1-2.

KLF9/13 sequences have been previously identified in many metazoan taxa but only the vertebrate clade is supported in our tree (figure 1c) [30,65]. Although not monophyletic, we hypothesize that the polytomies neighbouring the vertebrate clade are KLF9/13 which includes sequences that have been identified in previous phylogenies [30]. Interestingly, spiralians and *Nautilus* are represented, but no other cephalopod has a KLF9/13 sequence, suggesting a loss in Coleoid cephalopods. KLF10/11 was previously reported to be a vertebrate-specific gene, but with the addition of more protostome taxa, we identify a highly supported monophyletic KLF10/11 group, with a cephalopod-specific duplication (figure 1c). *D. melanogaster* does not possess a KLF10/11 gene, but orthologues are found in *Tribolium, Limulus, Daphnia*, and multiple spiralian sequences.

The SP family is strongly supported in the tree and the phylogeny shows two SP groups, SP1-4 and SP5-9, neither with strong support (figure 1c). Protostomes, including cephalopods, have one SP1-4 family member, with the exception of *D. pealeii*, which is unique in having two. SP5 and

SP6-9 groups were not resolved, unlike in other trees with less spiralian sampling. Spiralia have variable numbers of SP5-9 genes, typically between 3 and 4.

The addition of spiralian taxa to the KLF/SP gene family tree reveals a pattern of spiralian-specific gains and losses in nearly every major KLF/SP group. This complex evolutionary trajectory differs from the majority of ecdysozoans and basal deuterostomes which have not significantly expanded their KLF/SP repertoires (figure 1c), and may be responsible for difficulties in assigning orthology [30,66]. As is the case for other gene families with a very small highly conserved region, to better resolve this difficult tree, it is necessary to focus our inquiry on specific gene families and to narrow our phylogenetic sampling where full-length sequences may help. We decided to focus on the SP6-9 group as it has undergone a duplication event, *DpSP6-9a* and *DpSP6-9b*, and is enriched in the eye and optic lobe RNA-seq time course.

## (b) Phylogenetic analysis of the SP6-9 amino acid sequences in Spiralia

Our large KLF/SP tree did not have sufficient support to identify SP5 and SP6-9 as distinct groups. To confirm orthology of our cephalopod sequences, we generated spiralian-specific SP6-9 trees. This sub-family shares much more sequence identity within Spiralia, therefore, when possible, alignments of full-length amino acid sequences were made to construct ML and Bayesian trees (electronic supplementary material, figure S3). Our Bayesian tree resolved SP5 as a monophyletic group within SP6-9 but our ML tree shows support for an SP6-9 group to the exclusion of SP1-5 (electronic supplementary material, figure S3). Previous work supports the conclusion of our ML tree but we proceed with a cephalopod-specific tree to confirm the SP5 and SP6-9 clades (see section *e*). We recovered three cephalopod SP6-9 paralogs, although we did not find SP6-9c in the *D. pealeii* transcriptome. We discovered single orthologues in Brachiopoda, Gastrotricha, Nemertea, and aculiferan molluscs, two paralogs in Rotifera, Platyhelminthes, Bryozoa, Phoronida, Gastropoda, Scaphopoda, and Bivalvia, and one to three copies in Annelida. We find modest support for monophyletic clades of SP6-9 orthologues in each spiralian lineage, but no support for deeper phylogenetic relationships between SP6-9 orthologues.

## (c) Expression analysis of Krüppel-like factor/specificity protein homologues in *Doryteuthis pealeii*

Little is known about the function of the *KLF/SP* genes in the Spiralia, so we cloned and performed *in situ* hybridization studies in *D. pealeii* on multiple members of the family, including *KLF5*, *SP1-4*, and *SP5*. We analysed gene expression at stages 19, 21, 23, 25, and 27 in whole-mount (electronic supplementary material, figure S4–S6) [67].

*KLF5* is expressed in yolk nuclei, in a salt and pepper pattern across the cerebral ganglion, in arm IV (tentacle), and the dorsal gill at stage 19 through 23. Later in development (stage 25), *KLF5* is still found in the arms and expands to the funnel and mantle tissue. At stage 27, the expanded epidermal expression is detected (electronic supplementary material, figure S4). *SP1-4a* is found ubiquitously expressed from stage 19 until stage 27, consistent with vertebrate and arthropod expression (electronic

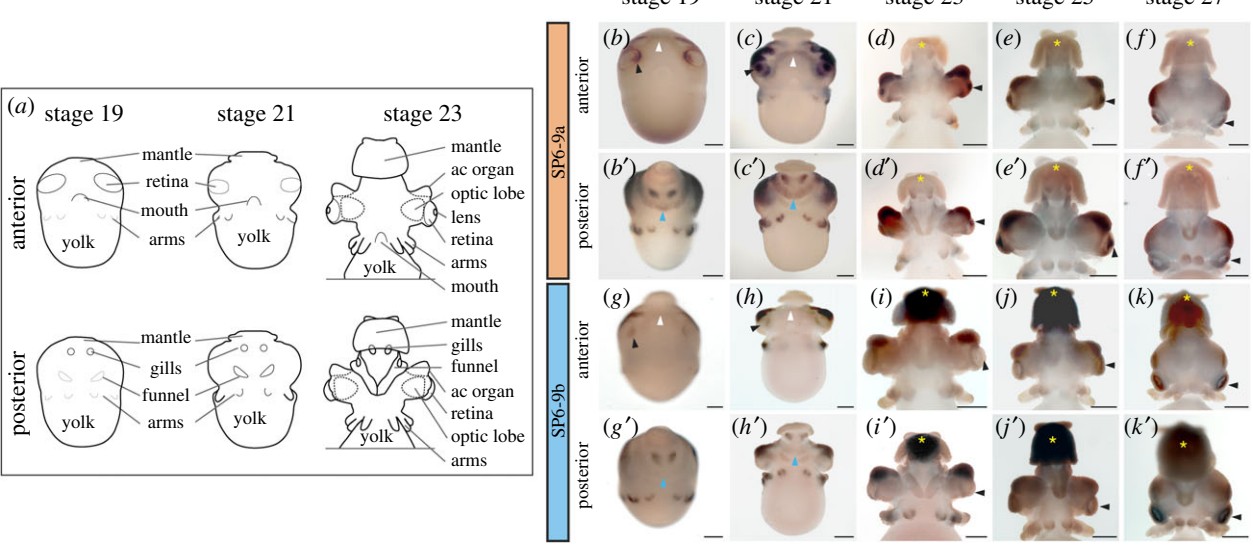

**Figure 2.** *DpSP6-9* paralog expression during development. (*a*) *D. pealeii* embryo schematics, anterior and posterior views. After stage 23, most organs remain recognizable in similar locations on the embryo. *In situ* hybridization of *DpSP6-9a* mRNA (*b–f'*) and *DpSP6-9b* mRNA (*g–k'*), stages 19 to 27. Expression is nearly identical for both genes in the nervous system, gills, arms, and anterior chamber organ. *DpSP6-9a* is highly expressed around the lens (black arrowheads), the cerebral ganglion (white arrowheads), and the palliovisceral ganglion (cyan arrowheads). *DpSP6-9b* expression domain is similar but lower in these corresponding areas (arrowheads), and differentially highly expressed in the mantle (yellow asterisks). Scale bars, 200 μm; ac organ, anterior chamber organ.

supplementary material, figure S5) [29] reviewed in [68]. The second member of the SP gene family assessed, *SP5*, has dynamic expression in *D. pealeii* including high expression in the optic lobe at stage 23 (electronic supplementary material, figure S6). In all stages, low expression is seen in the retina, and anterior segment expression is seen at stage 27.

## (d) Paralog-specific expression of the *DpSP6-9* genes

To understand the consequence of gene duplication, we also performed expression studies for both *DpSP6-9* paralogs. *DpSP6-9a* and *DpSP6-9b* have significant overlap in their expression with notable differences. Both *DpSP6-9a* and *DpSP6-9b* show expression on the medial side of the placode at stages 19 and 21 (figures 2*b,c,g,h* and 3*a–i*). The expression is also seen in the gills and the limbs in *D. pealeii* (figures 2 and 3). SP6-9 is part of a limb gene regulatory network found in vertebrates and ecdysozoans [69–75]. Recent findings have shown *SP6-9a* (*SP8/9a*) expressed early in cuttlefish limb outgrowth [12]. Both paralogs are also expressed in the anterior chamber organ from stage 19 through to stage 27, a tissue currently hypothesized to control ocular pressure but without a well-understood function [76].

Despite their similar expression patterns, these paralogs do show differences in their domains. *DpSP6-9a* shows enriched expression in regions hypothesized to form the cerebral ganglion and the palliovisceral ganglion (figure 2*c*) and *DpSP6-9b* shows unique expression in the mantle at stage 23 until hatching (figure 2*i–k*). Major differences in expression are found in the developing eye. At stage 19, the expression of *DpSP6-9a* extends in all lip cells around the developing placode, while *DpSP6-9b* expression is restricted from the lip, with the exception of the most medial cells (figure 3*a–e*). At stage 21, the lip cells have fused in the anterior of the optic vesicle. At this stage, *DpSP6-9a* is expressed in the anterior segment and *DpSP6-9b* is significantly reduced in the anterior

segment (figure 3*a,f–i*). This difference in expression is maintained through to stage 27. At stages close to hatching, *DpSP6-9a* is in the tertiary lentigenic cells in the anterior segment (figure 3*t*) [39,40].

This significant difference in expression suggests that SP6-9a plays a role in lens development while SP6-9b does not. Vertebrate and *Drosophila* homologues of SP6-9 do not have an association with visual system development but do show expression in the central nervous system [75,77,78]. However, SP6-9 expression in the retina may be ancestral in spiralian visual systems as SP6-9 is required for eye cup regeneration in the flatworm *Schmidtea mediterranea* in conjunction with the transcription factor Dlx [79]. The evolution of novel expression domains is characteristic of neofunctionalization after duplication [80]. In finding this paralog-specific expression correlated with a cephalopod novelty (the lens), we wanted to further investigate this duplication event.

## (e) Cephalopod SP6-9 sequence evolution and synteny

To understand the molecular changes in the cephalopod SP6-9 genes correlated with divergent expression and to confirm our SP6-9/SP5 orthology, we constructed a Bayesian tree with only cephalopod members (figure 4*a*). This tree shows support for separate SP5 and SP6-9 groups. It also shows SP6-9a and SP6-9b as sister groups to the exclusion of SP6-9c in all cephalopods. This relationship between paralogs includes *Nautilus*, which do not have a lens. In addition, a long branch leads to cephalopod SP6-9a, but within this group sequences are highly similar. This contrasts with relatively shorter branches leading to both SP6-9b and SP6-9c clades, but increased sequence divergence within each group across cephalopod lineages (figure 4*a*). The similarity of cephalopod SP6-9a sequences includes the lensless *Nautilus*, which does not express any *SP6-9* homologue in its eye [43]. Together this suggests that coding sequence evolution has not altered the

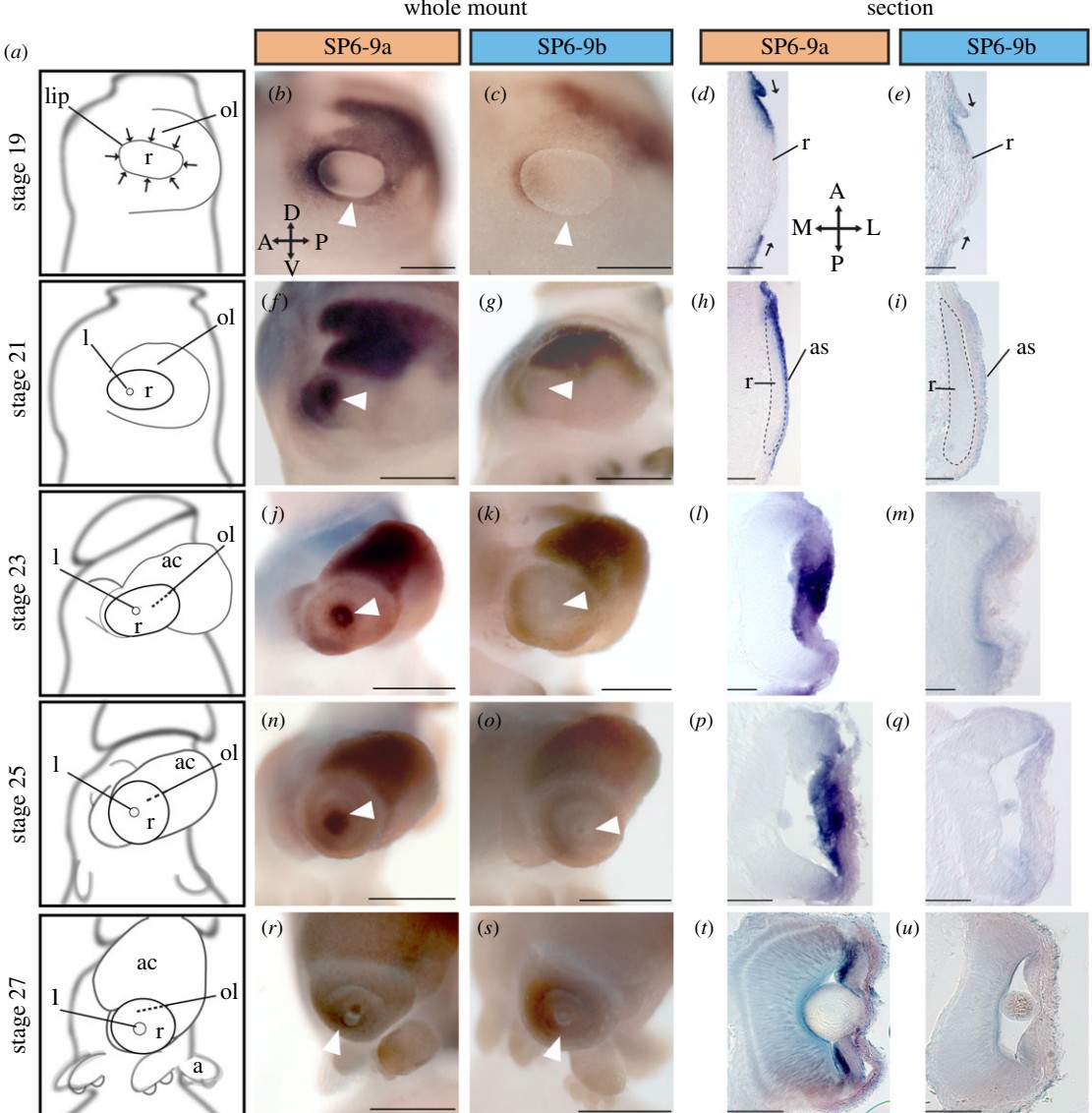

**Figure 3.** *DpSP6-9a* but not *DpSP6-9b* is highly expressed in lentigenic cells of the developing eye. (*a*) Schematics showing lateral views of the developing eye, anterior is left. At stage 19, the retina placode is exposed and being internalized by a surrounding lip of cells (arrows). The vesicle is closed at stage 21 and the lip cells generate the lens and anterior segment. ol, optic lobe; lip, lip of cells that internalizes the retinal placode; l, lens, r, retina; ac, anterior chamber organ; a, arms. (*b–u*) Lateral view and cryosections of *DpSP6-9a* and *DpSP6-9b*. White arrowheads highlight the region of lip and developing lentigenic cells. In sections, anterior of the animal is up, medial is left. Expression in the retinal placode can be seen for both genes, r, retina. At stage 19, *DpSP6-9a* is highly expressed on both sides of the lip as it closes (arrows), where *DpSP6-9b* is restricted medially (*b–e*). At stage 21, the anterior of the eye vesicle expresses *DpSP6-9a* with minimal expression of *DpSp6-9b* r, retina; as, anterior segment (*f–i*). Later *DpSP6-9a* is strongly expressed in the lentigenic cells. Again *DpSP6-9b* is reduced (*j–u*). Asterisk, panel (*t*) is a stage 28 embryo; scale bars, 200 μm.

ancestral molecular function of *SP6-9a*. Instead, *SP6-9a* may perform a similar biochemical function among cephalopods, but its novel recruitment to the anterior segment is likely a result of a *cis*-regulatory change. Wnt is a well-studied regulator of *SP6-9* homologues in both vertebrates and arthropods, and may play a role regulating this new expression domain in cephalopods [72,73,81–83].

Defining relationships among *SP6-9* orthologues and paralogs is difficult with phylogenetic methods alone, so we sought additional evidence to assess their evolutionary history. We evaluated gene architecture and synteny in spiralian *SP6-9* paralogs (figure 4*b*, electronic supplementary material, table S3). We found multiple instances of spiralian *SP6-9* genes that were intronless, including all cephalopod *SP6-9b* and *SP6-9c* sequences. In addition, cephalopods share synteny across all SP6-9 paralogs. This includes *SP6-9b*, *SP6-9c*, and

*SP6-9a* in sequential order, in the same 5′ to 3′ direction, within 1.5 Mb of each other in all cephalopod genomes investigated so far. We also find that *SP6-9a* is the only paralog in cephalopods found to have introns. *SP6-9* duplicates in other spiralians are on the same chromosome, but do not show the obvious synteny found within cephalopods (figure 4*b*). These tandem, intronless paralogs suggest two duplication events via retrotransposition early in the cephalopod lineage and maintained in the genome [84]. Retrotransposition is an important contributor to insertion events and structural variation in both vertebrate and arthropod species [85,86]. This may be an unusual case in cephalopods; however, despite the fact that the octopus genome is enriched in transposon activity, recent analyses have shown retrotransposon-derived genomic elements only make up 2–8% of molluscan genomes, as opposed to 35–52% in vertebrates [10,51,87–89].

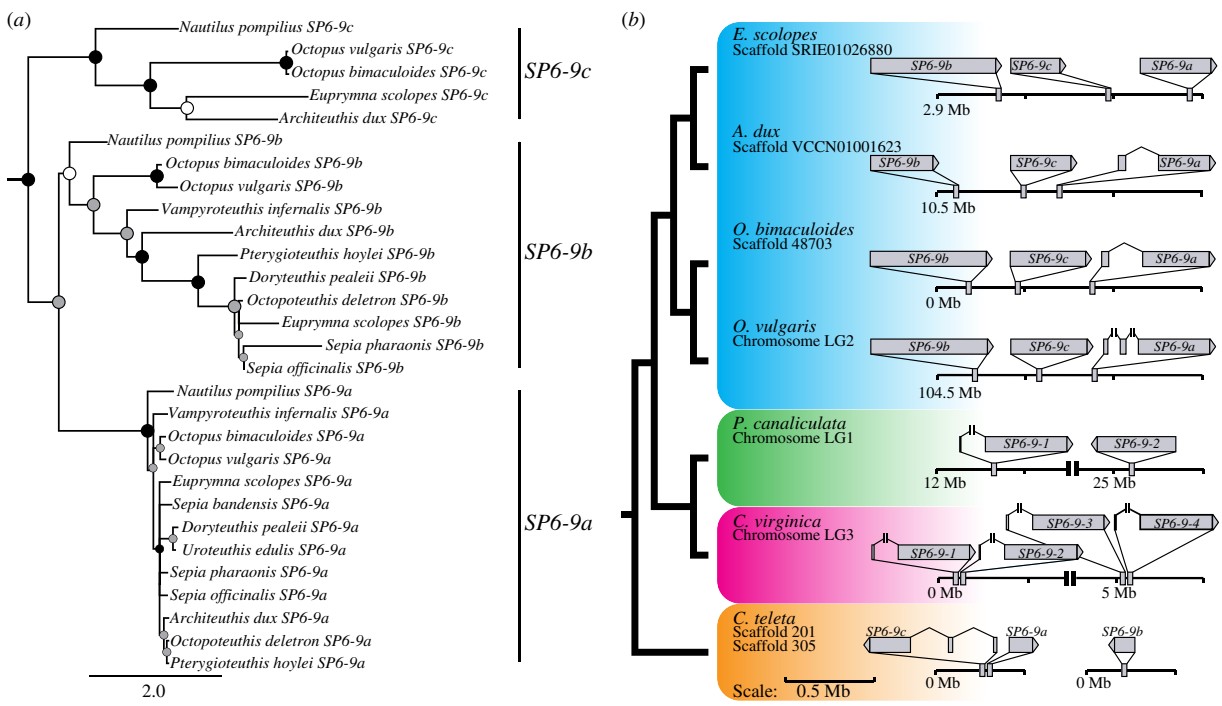

**Figure 4.** Synteny and cephalopod-specific phylogeny reveal *SP6-9* evolutionary history. (*a*) Bayesian tree shows support for *SP6-9c* as the outgroup to *SP6-9a* and b in Cephalopoda. Branch lengths show divergence in *SP6-9b* and c while little sequence change has occurred within the *SP6-9a* clade across cephalopods. Circles represent posterior probabilities above 0.5 (white), 0.7 (grey), and 0.95 (black). (*b*) Mapping *SP6-9* paralog genomic locations shows cephalopod genomic architecture is distinct from other spiralians. Cephalopod *SP6-9a* genes have introns while *SP6-9b* and *SP6-9c* do not. The four paralogs in *Crassostrea virginica* may represent a recent genomic duplication or genome assembly error. Genomic backbones in black are all to the same scale, and zoomed-in gene lengths are all to scale with one another, except where breaks are indicated with double vertical bars. Taxon colours correspond to colours used in the trees in figure 5 and electronic supplementary material figure S3.

## (f) Character mapping duplications and losses of SP6-9

From our SP6-9 trees, assigning specific lineage duplications and losses is difficult without support from a species phylogeny. We wanted to clarify if these SP6-9 duplications were cephalopod-specific and therefore needed to better understand the relationship of SP6-9 paralogs across Spiralia. We first estimated the number of SP6-9 duplicates at the base of Mollusca and Annelida. Our spiralian SP6-9 trees did not give strong support for deeper nodes, so we assembled additional transcriptomes with a particular focus on basal branching annelid and molluscan lineages, and generated Bayesian SP6-9 trees specific to each group (electronic supplementary material, figures S7 and S8). We mapped the most parsimonious patterns of duplication and loss onto the current species tree [52–57,90,91] (figure 5*a*). We infer that the ancestral spiralian had a single SP6-9 orthologue and multiple independent duplications occurred in Rotifera, Platyhelminthes, Annelida, Mollusca, and the lineage leading to Phoronida & Bryozoa.

Within annelids, the Serpulidae, Clitellata, Apitellidae, and Echiura have three paralogs while basally branching members of Sedentaria, Sabellidae, and Orbiniidae, as well as all other annelid groups have one (figure 5*b*, electronic supplementary material, figure S3). Within Sedentaria, we do not find support for 1 : 1 orthologues of SP6-9 sequences. Instead two paralogs of Clitellata are supported in a clade with a single Echiura/Capitellidae/Serpulidae orthologue, while another SP6-9 clade contains two Echiura/Capitellidae/Serpulidae paralogs and a single Clitellata orthologue. This pattern suggests two duplications at the base of the Sedentaria lineage followed by a loss and then another duplication in Clitellata (figure 5*b*, electronic supplementary material, figures S3 and S7).

Within molluscs, the most parsimonious hypothesis supports a duplication at the base of Conchifera and a second duplication in Cephalopoda (figure 5*c*) [52,54,56,57]. Aculiferan species had either one SP6-9 homologue, or no SP6-9 transcript was found (electronic supplementary material, tables S2, S4 and figure S8). All bivalves and gastropods have two paralogs with the exception of Protobranchia which has one SP6-9 orthologue. This supports a lineage-specific loss, with the understanding that transcriptomic sampling may be incomplete (figure 5*c*, electronic supplementary material, figure S3, pink). Within Gastropoda, the Hypsogastropoda have two SP6-9 paralogs but they fall within a single well-supported gastropod SP6-9 clade (electronic supplementary material, figure S3). Genomic evidence confirms the loss of one gastropod paralog and duplication of gastropod SP6-9-1 within the Hypsogastopoda lineage (figure 5*c*, electronic supplementary material, figure S3, green). The cephalopod-specific SP6-9 duplication and its relationship with a novel lens sit within a broader dynamic pattern of gene gain and loss in Spiralia. Ultimately, further research in this major branch of the animal tree will shed light on whether these patterns are in part responsible for the vast morphological diversity found in this group.

## 4. Conclusion

The cephalopod eye and optic lobe together is a remarkable example of a sophisticated biological system and the lens is one of its most apparent novelties. In this study, we identified at least one cephalopod-specific gene duplication of the SP6-9

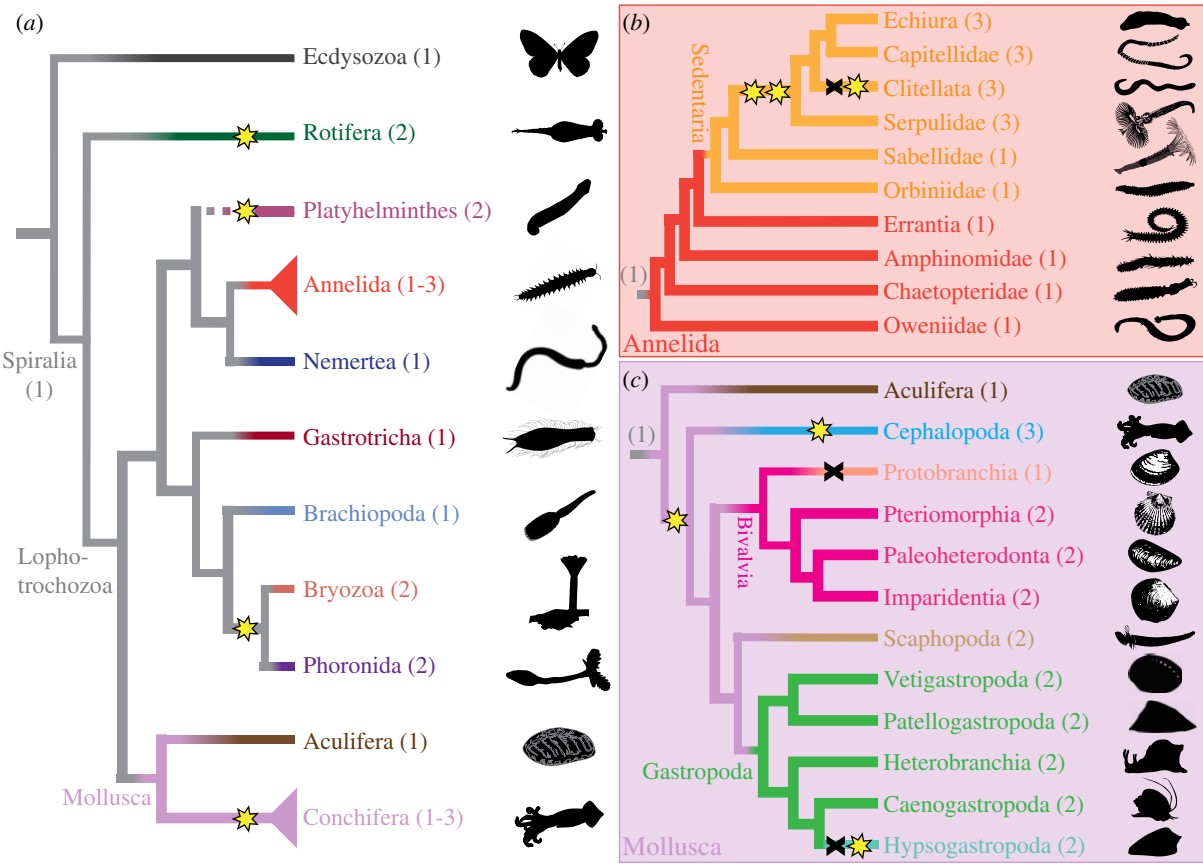

**Figure 5.** SP6-9 has a complex pattern of duplication across Spiralia and Cephalopoda. (*a*) The most parsimonious pattern of duplication events mapped onto a current spiralian phylogeny. At minimum, separate duplications occurred in Rotifera, Platyhelminthes, Annelida, the branch leading to Phoronida & Bryozoa, and Conchifera. Annelida and Mollusca subtrees are collapsed and detailed in (*b*) and (*c*). (*b*) Annelida: amino acid tree evidence indicates two duplications within Sedentaria after the split of Orbiniidae and Sabellidae, followed by a loss and subsequent duplication event in Clitellata. Basally branching annelids have a single paralog only, suggesting a single paralog at the base of Annelida. (*c*) Mollusca: this hypothesis suggests a duplication event following the split of Aculifera. At minimum, there was a duplication in Cephalopoda, a loss in Protobranchia and a loss and duplication in Hypsogastropoda. Phylogenies based on [52–61]. In all panels: star, gene duplication; X, gene loss; images from Phylopic.org, or drawn by K. McCulloch, with credit for *Capitella* and *Platynereis* B. Duygu Özpolat, and credit for abalone and owl limpet to Taro Maeda.

gene as supported by similar gene expression, synteny, gene architecture, and phylogenetics. SP6-9a differs from SP6-9b in its robust expression in the lentigentic cells in the developing cephalopod anterior segment. This is the first study connecting a cephalopod-specific gene duplication to a visual system neofunctionalization. It opens the door to many mechanisms for elaboration: the evolution of new lens-specific *cis*-regulatory targets, or even the redeployment of canonical regulatory networks in the anterior segment of the cephalopod eye. This study highlights the exciting opportunities within Spiralia to address fundamental questions underlying the evolution of novel phenotypes, including the consequences of gene duplication and loss, as well as changes in *cis*-regulation.

**Ethics.** The animals were treated in accordance with the European directive 86/609/EEC (D.Lgs.n. 26/2014) with the goal of replacement, reduction, refinement, and the minimization of animal suffering.

**Data accessibility.** Cloning primers, alignment files, tree files with branch lengths, species list, sequence accession numbers, and references supporting our phylogenetic dataset have been uploaded as part of the electronic supplementary material. Two full-length and nine partial *Doryteuthis pealeii* KLF/SP coding sequences are deposited in Genbank with accession numbers MT118824–MT118834.

**Authors' contributions.** K.M.K. and K.J.M. designed all the experiments. K.J.M. and K.M.K. performed experiments. K.J.M. performed phylogenetic analyses. K.M.K. and K.J.M. wrote the manuscript.

**Competing interests.** We declare we have no competing interests.

**Funding.** K.M.K. and K.J.M.'s work is supported by the Office of the NIH Director 1DP5OD023111-01, Whitman Center Research Award, Baxter Postdoctoral Fellowship Fund, Eugene and Milicent Bell Fellowship Fund in Tissue Engineering at the MBL, the Elisabet Samuelsson Director's Discretionary Fund, and Harvard University.

**Acknowledgements.** We would like to thank the Koenig and Srivastava lab members for helpful discussions. We also acknowledge the support of the John Harvard Distinguished Science Fellows community. We also thank the MBL and the Marine Resources Center.

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
