## [Reviewer comments · Proceedings of the Royal Society B: Biological Sciences]

Review History

RSPB-2020-0734.R0 (Original submission)

Review form: Reviewer 1

Recommendation

Major revision is needed (please make suggestions in comments)

Scientific importance: Is the manuscript an original and important contribution to its field?

Acceptable

General interest: Is the paper of sufficient general interest?

Acceptable

Quality of the paper: Is the overall quality of the paper suitable?

Acceptable

Is the length of the paper justified?

Yes

Should the paper be seen by a specialist statistical reviewer?

No

Do you have any concerns about statistical analyses in this paper? If so, please specify them explicitly in your report.

No

It is a condition of publication that authors make their supporting data, code and materials available - either as supplementary material or hosted in an external repository. Please rate, if applicable, the supporting data on the following criteria.

Is it accessible?

No

Is it clear?

Yes

Is it adequate?

Yes

Do you have any ethical concerns with this paper?

Yes

Comments to the Author

See attached document. (See Appendix A)

Review form: Reviewer 2 (Tim Wollesen)

Recommendation

Major revision is needed (please make suggestions in comments)

Scientific importance: Is the manuscript an original and important contribution to its field?

Excellent

General interest: Is the paper of sufficient general interest?

Good

Quality of the paper: Is the overall quality of the paper suitable?

Good

Is the length of the paper justified?

Yes

Should the paper be seen by a specialist statistical reviewer?

No

Do you have any concerns about statistical analyses in this paper? If so, please specify them explicitly in your report.

No

It is a condition of publication that authors make their supporting data, code and materials available - either as supplementary material or hosted in an external repository. Please rate, if applicable, the supporting data on the following criteria.

Is it accessible?

Yes

Is it clear?

Yes

Is it adequate?

Yes

Do you have any ethical concerns with this paper?

No

Comments to the Author

McCulloch and Koenig study the expression, orthology, and synteny of members of the KLF/SP gene family in the Spiralia with special focus on the cephalopod *Doryteuthis pealeii*. Their work follows up on aspects of a previous study which revealed an enriched expression of KLF/SP genes in the developing eyes of squid (König et al. 2016). In the present manuscript the authors mined the transcriptome of cephalopods as well as other spiralian for members of this gene family. The authors performed in situ hybridization experiments on several members of this family in squid developmental stages. Using published genomes and transcriptomes they conclude that the individual spiralian clades have divergent numbers of SP6-9 paralogs which show synteny in cephalopods but not the other spiralian. Among the squid paralogs, SP6-9a is expressed in the squid lens in contrast to SP6-9b leading the authors to infer that gene duplication led to neofunctionalization of this paralog.

The study is well-conceived with convincing gene expression patterns and sound phylogenetic analyses. It contributes to our understanding on the evolution of coleoid cephalopod complex eyes and poses many more questions such as how these genes might be functional in *Nautilus* or the eyeless scaphopods.

I have few points of criticism that should be easily addressed:

Major issues:

a.) Some clades are definitively underrepresented in the ancestral state reconstruction shown in Fig.5b, c. In particular, basal-branching taxa such as the aculiferan Caudofoveata and Solenogastres are not included although transcriptomes do exist. There should also be publicly available transcriptomes of basally branching annelids such as *Owenia* and *Chaetopterus* (check work by C. Bleidorn and C. Helm for phylogeny and possibly transcriptomes). It is important to include these taxa since they may change the outcome of the analysis.

The authors state “branches for which we have data” (l. 139). I have to admit that it is easy to miss out on new published transcriptomes, but the authors should definitively include the above-mentioned ones. If no orthologs are found, then this should be stated in the figure legend.

b.) Why do the authors argue for one SP6-9 ortholog instead of two paralogs in the last common molluscan ancestor? In my opinion, an evolutionary scenario favoring two paralogs would be more parsimonious since less changes would be necessary? Maybe I overread some arguments in the manuscript?

c.) The authors barely cite references and their own beautiful figures in their manuscript (see below).

d.) The figure (legends) should be self-explaining: Often gene expression domains are not labeled properly in the figures and labels are not explained in the legends (see below).

Minor issues:

l. 28: single-chambered eyes

l. 37: which other species? provide references.

l. 45: I reckon it might be important to go a little more into detail about the function of these genes in non-spiralian taxa (only few examples).

l. 49: “developing eye and optic lobe tissue”: replace “tissue” with “tissues”. Here it would be important to provide an explanation for laypersons who are not familiar with the cephalopod nervous system (“optic lobe”).

l. 50: same here, please explain “Coleoid”. It should also be written lowercase. A reference for “significant innovations...” would be useful.

- l. 53: here I would write coleoid cephalopods, since - as the authors stated - nautilus have different eyes ("pinhole-type eyes")
- l. 56. Here references are missing - check throughout manuscript
- l. 64. I would recommend including a sketch drawing of the eyes including all those layers, lens etc.
- l. 70: This "cephalopod duplications" sounds awkward, better "we identify duplications of the KLF/SP gene families in cephalopods..."
- l. 77: replace "these studies" with "our study"
- l. 84: "kept in flowing seawater" - same water temperature?
- l. 89: "PCR products were cloned", but size-fractionated before, right?
- l. 117: "identifiedSP6-9" space missing
- l. 133 "Annelids which lack a chromosome-level assembly.": Reference is missing and Annelids should be written in lower-case.
- l. 137: "current hypotheses of relationships": add "phylogenetic"
- l. 138: annelids, spiralian, molluscs, BUT Spiralia, Annelida, Mollusca... Please correct throughout manuscript
- l. 156: Reference missing
- l. 158: "To produce the alignment..." this sentence should be part of the M&M
- l. 180: metazoan, not Metazoan
- l. 181: ref missing
- l. 186: cite Figure
- l. 186: I would rephrase the sentence "D. melanogaster does not have a 187 member of KLF10/11, but we do find Tribolium, Limulus, Daphnia, and multiple spiralian 188 sequences." to "D. melanogaster does not possess a KLF10/11 gene but orthologs are found in ..."
- l. 190: cite Figure
- l. 214: I would refer to polyplacophoran mollusks since you did not introduce polyplacophorans before (not everyone in the audience is familiar with the individual molluscan clades). Same with scaphopods.
- l. 222: sounds like Arnold did the in situ, please refer also to your figure ;)
- l. 225: how do you define early development? Better mention stages, at least in parenthesis.
- l. 225: "Arm and mantle expression is apparent later" - that contradicts the previous sentence (you already mentioned "arm" development)
- Fig. S3: expressed in cells covering the yolk sac. Inside the yolk sac as well?
- l. 229: Why do the authors discuss SP5 expression here again? I would reorganize this paragraph accordingly.
- l. 237: Citation of Fig. is misleading since you also cite stages stage 24 and 27: Better cite "Fig. 2b-c, g-h..."
- l. 237: better: SP6-9...expression in retina may be
- l. 240: reference missing
- l. 240-242: A role in what? I would combine both sentences and write "...limp outgrowth and our data suggest that both paralogs of SP6-9 may play a similar role"
- l. 246: "Despite their similar expression, these paralogs do show differences." In what? Better phrase differently such as: "despite shared expression domains, these paralogs are also expressed in different regions.
- l. 248: "The most dramatic", I would replace it with "Major differences"
- l. 247: Why do you assume that these expression domains correspond to the cerebral and palliovisceral ganglia, respectively? I would expect the cerebral ganglia to be located more anteriorly and not that dorsal during this early dev. stage. Do you have any neural markers that would support your notion?
- Also, please label both expression domains with differently colored arrows/ other labels.
- l. 273: ...is correlated with coleoid lens evolution.
- l. 281: in all cephalopod genomes investigated so far.
- l.290: "This hypothesis suggests", better "we hypothesize.." or "we infer"
- l. 291: This could be misunderstood. Better: "...Annelida, Mollusca, and the lineage leading to Phoronida and Bryozoa".

Please also change "Chonchifera" to "Conchifera"

l. 296: Clitellata

l. 300: Why did the authors not include basally-branching annelids such as *Owenia* and *Chaetopterus* (see above)?

Figure legend 5: It is important to cite the studies of the phylogenetic trees on which the characters are plotted.

l. 302: Please add references with respect to the phyl. studies (see above) e.g. (see Kocot et al. 2011, Smith et al. 2011?).

l.302: I am not entirely happy with the molluscan tree and the annelid tree since basally-branching taxa are underrepresented (see above). There are other aculiferan transcriptomes than the polyplacophoran one, e.g. of solenogastres and caudofoveates (De Oliveira AL, Wollesen T, Kristof A, Scherholz M, Redl E, Todt C, Bleidorn C, Wanninger A. 2016. Comparative transcriptomics enlarges the toolkit of known developmental genes in mollusks. *BMC Genomics* 17: 905 (23 pages). DOI: 10.1186/s12864-016-3080-9).

An increased number of paralogs (e.g. 2) among the Aculifera would change the phylogenetic scenario. Then polyplacophorans would have probably lost a paralog and a duplication event in the conchiferan stem lineage would have been unlikely. A similar scenario could also be possible for the annelid tree.

l.303: Conchifera (see also Fig. 5)

l. 303: Cite Figure

l. 320: "in the anterior segment." Specify of what?

Discussion

The authors state in the introduction that KLF/SP proteins have "diverse developmental, cellular and homeostatic" functions. Are these genes also implicated in eye development in other organisms? If yes, it would be worth discussing.

Figures:

Although the authors provide a sketch drawing depicting Dp developmental stages, it is necessary to label all structures in the micrographs that express a given gene (e.g. gills and arms in Figure S3), otherwise it is too cumbersome when reading the text body and checking the Figures.

Fig. 4a: Explain branch support (filled grey, black, white circles)

Fig. 5:

- 5c: What's the difference between a black and a grey "X" (gene loss). Should be indicated in the legend.

- 5a: How many paralogs did the last common spiralian ancestor have?

Decision letter (RSPB-2020-0734.R0)

05-Jun-2020

Dear Dr Koenig:

I am writing to inform you that your manuscript RSPB-2020-0734 entitled "KLF/SP Transcription Factor Evolution in the Spiralia and the Implications for Cephalopod Visual System Novelty" has, in its current form, been rejected for publication in *Proceedings B*.

This action has been taken on the advice of referees, who have recommended that substantial revisions are necessary. With this in mind we would be happy to consider a resubmission,

provided the comments of the referees are fully addressed. However please note that this is not a provisional acceptance.

Sincerely,
Dr Sasha Dall
<mailto:proceedingsb@royalsociety.org>

Associate Editor

Board Member: 1

Comments to Author:

Two experts in the field have reviewed your manuscript, and they agree on the general relevance of the work. However, they have identified some significant issues (e.g. including the overall structure of the manuscript and the interpretation of the results). As a consequence of the comments, I cannot recommend the MS for publication.

Reviewer(s)' Comments to Author:

Referee: 1

Comments to the Author(s)

See attached document

Referee: 2

Comments to the Author(s)

McCulloch and Koenig study the expression, orthology, and synteny of members of the KLF/SP gene family in the Spiralia with special focus on the cephalopod *Doryteuthis pealeii*. Their work follows up on aspects of a previous study which revealed an enriched expression of KLF/SP genes in the developing eyes of squid (König et al. 2016). In the present manuscript the authors mined the transcriptome of cephalopods as well as other spiralian for members of this gene family. The authors performed in situ hybridization experiments on several members of this family in squid developmental stages. Using published genomes and transcriptomes they conclude that the individual spiralian clades have divergent numbers of SP6-9 paralogs which

show synteny in cephalopods but not the other spiralian. Among the squid paralogs, SP6-9a is expressed in the squid lens in contrast to SP6-9b leading the authors to infer that gene duplication led to neofunctionalization of this paralog.

The study is well-conceived with convincing gene expression patterns and sound phylogenetic analyses. It contributes to our understanding on the evolution of coleoid cephalopod complex eyes and poses many more questions such as how these genes might be functional in *Nautilus* or the eyeless scaphopods.

I have few points of criticism that should be easily addressed:

Major issues:

- a.) Some clades are definitively underrepresented in the ancestral state reconstruction shown in Fig.5b, c. In particular, basal-branching taxa such as the aculiferan Caudofoveata and Solenogastres are not included although transcriptomes do exist. There should also be publicly available transcriptomes of basally branching annelids such as *Owenia* and *Chaetopterus* (check work by C. Bleidorn and C. Helm for phylogeny and possibly transcriptomes). It is important to include these taxa since they may change the outcome of the analysis. The authors state “branches for which we have data” (l. 139). I have to admit that it is easy to miss out on new published transcriptomes, but the authors should definitively include the above-mentioned ones. If no orthologs are found, then this should be stated in the figure legend.
- b.) Why do the authors argue for one SP6-9 ortholog instead of two paralogs in the last common molluscan ancestor? In my opinion, an evolutionary scenario favoring two paralogs would be more parsimonious since less changes would be necessary? Maybe I overread some arguments in the manuscript?
- c.) The authors barely cite references and their own beautiful figures in their manuscript (see below).
- d.) The figure (legends) should be self-explaining: Often gene expression domains are not labeled properly in the figures and labels are not explained in the legends (see below).

Minor issues:

- l. 28: single-chambered eyes
- l. 37: which other species? provide references.
- l. 45: I reckon it might be important to go a little more into detail about the function of these genes in non-spiralian taxa (only few examples).
- l. 49: “developing eye and optic lobe tissue”: replace “tissue” with “tissues”. Here it would be important to provide an explanation for laypersons who are not familiar with the cephalopod nervous system (“optic lobe”).
- l. 50: same here, please explain “Coleoid”. It should also be written lowercase. A reference for “significant innovations...” would be useful.
- l. 53: here I would write coleoid cephalopods, since - as the authors stated - nautilus have different eyes (“pinhole-type eyes”).
- l. 56. Here references are missing – check throughout manuscript
- l. 64. I would recommend including a sketch drawing of the eyes including all those layers, lens etc.
- l. 70: This “cephalopod duplications” sounds awkward, better “we identify duplications of the KLF/SP gene families in cephalopods...”
- l. 77: replace “these studies” with “our study”
- l. 84: “kept in flowing seawater” – same water temperature?
- l. 89: “PCR products were cloned”, but size-fractionated before, right?
- l. 117: “identifiedSP6-9” space missing
- l. 133 “Annelids which lack a chromosome-level assembly.”: Reference is missing and Annelids should be written in lower-case.
- l. 137: “current hypotheses of relationships”: add “phylogenetic”
- l. 138: annelids, spiralian, molluscs, BUT Spiralia, Annelida, Mollusca... Please correct throughout manuscript
- l. 156: Reference missing
- l. 158: “To produce the alignment...” this sentence should be part of the M&M
- l. 180: metazoan, not Metazoan

- l. 181: ref missing
- l. 186: cite Figure
- l. 186: I would rephrase the sentence "D. melanogaster does not have a 187 member of KLF10/11, but we do find Tribolium, Limulus, Daphnia, and multiple spiralian 188 sequences." to "D. melanogaster does not possess a KLF10/11 gene but orthologs are found in ..."
- l. 190: cite Figure
- l. 214: I would refer to polyplacophoran mollusks since you did not introduce polyplacophorans before (not everyone in the audience is familiar with the individual molluscan clades). Same with scaphopods.
- l. 222: sounds like Arnold did the in situ, please refer also to your figure ;)
- l. 225: how do you define early development? Better mention stages, at least in parenthesis.
- l. 225: "Arm and mantle expression is apparent later" – that contradicts the previous sentence (you already mentioned "arm" development)
- Fig. S3: expressed in cells covering the yolk sac. Inside the yolk sac as well?
- l. 229: Why do the authors discuss SP5 expression here again? I would reorganize this paragraph accordingly.
- l. 237: Citation of Fig. is misleading since you also cite stages stage 24 and 27: Better cite "Fig. 2b-c, g-h..."
- l. 237: better: SP6-9...expression in retina may be
- l. 240: reference missing
- l. 240-242: A role in what? I would combine both sentences and write "...limp outgrowth and our data suggest that both paralogs of SP6-9 may play a similar role"
- l. 246: "Despite their similar expression, these paralogs do show differences." In what? Better phrase differently such as: "despite shared expression domains, these paralogs are also expressed in different regions."
- l. 248: "The most dramatic", I would replace it with "Major differences"
- l. 247: Why do you assume that these expression domains correspond to the cerebral and palliovisceral ganglia, respectively? I would expect the cerebral ganglia to be located more anteriorly and not that dorsal during this early dev. stage. Do you have any neural markers that would support your notion?
- Also, please label both expression domains with differently colored arrows/ other labels.
- l. 273: ...is correlated with coleoid lens evolution.
- l. 281: in all cephalopod genomes investigated so far.
- l.290: "This hypothesis suggests", better "we hypothesize.." or "we infer"
- l. 291: This could be misunderstood. Better: "...Annelida, Mollusca, and the lineage leading to Phoronida and Bryozoa".
- Please also change "Chonchifera" to "Conchifera"
- l. 296: Clitellata
- l. 300: Why did the authors not include basally-branching annelids such as Owenia and Chaetopterus (see above)?
- Figure legend 5: It is important to cite the studies of the phylogenetic trees on which the characters are plotted.
- l. 302: Please add references with respect to the phyl. studies (see above) e.g. (see Kocot et al. 2011, Smith et al. 2011?).
- l.302: I am not entirely happy with the molluscan tree and the annelid tree since basally-branching taxa are underrepresented (see above). There are other aculiferan transcriptomes than the polyplacophoran one, e.g. of solenogastres and caudofoveates (De Oliveira AL, Wollesen T, Kristof A, Scherholz M, Redl E, Todt C, Bleidorn C, Wanninger A. 2016. Comparative transcriptomics enlarges the toolkit of known developmental genes in mollusks. BMC Genomics 17: 905 (23 pages). DOI: 10.1186/s12864-016-3080-9).
- An increased number of paralogs (e.g. 2) among the Aculifera would change the phylogenetic scenario. Then polyplacophorans would have probably lost a paralog and a duplication event in the conchiferan stem lineage would have been unlikely. A similar scenario could also be possible for the annelid tree.
- l.303: Conchifera (see also Fig. 5)

l. 303: Cite Figure

l. 320: "in the anterior segment." Specify of what?

Discussion

The authors state in the introduction that KLF/SP proteins have "diverse developmental, cellular and homeostatic" functions. Are these genes also implicated in eye development in other organisms? If yes, it would be worth discussing.

Figures:

Although the authors provide a sketch drawing depicting Dp developmental stages, it is necessary to label all structures in the micrographs that express a given gene (e.g. gills and arms in Figure S3), otherwise it is too cumbersome when reading the text body and checking the Figures.

Fig. 4a: Explain branch support (filled grey, black, white circles)

Fig. 5:

- 5c: What's the difference between a black and a grey "X" (gene loss). Should be indicated in the legend.

- 5a: How many paralogs did the last common spiralian ancestor have?

Author's Response to Decision Letter for (RSPB-2020-0734.R0)

See Appendix A.

RSPB-2020-2055.R0

Review form: Reviewer 2 (Tim Wollesen)

Recommendation

Accept as is

Scientific importance: Is the manuscript an original and important contribution to its field?

Excellent

General interest: Is the paper of sufficient general interest?

Excellent

Quality of the paper: Is the overall quality of the paper suitable?

Excellent

Is the length of the paper justified?

Yes

Should the paper be seen by a specialist statistical reviewer?

No

Do you have any concerns about statistical analyses in this paper? If so, please specify them explicitly in your report.

No

It is a condition of publication that authors make their supporting data, code and materials available - either as supplementary material or hosted in an external repository. Please rate, if applicable, the supporting data on the following criteria.

Is it accessible?

Yes

Is it clear?

Yes

Is it adequate?

Yes

Do you have any ethical concerns with this paper?

Yes

Comments to the Author

Dear authors,

Congratulations on your interesting manuscript. You made a big effort to address all points of criticism and to improve the quality of your manuscript.

Best,
Tim Wollesen

Decision letter (RSPB-2020-2055.R0)

18-Sep-2020

Dear Dr Koenig

I am pleased to inform you that your Review manuscript RSPB-2020-2055 entitled "KLF/SP Transcription Factor Evolution in the Spiralia and the Implications for Cephalopod Visual System Novelties" has been accepted for publication in Proceedings B.

The referee(s) do not recommend any further changes. Therefore, please proof-read your manuscript carefully and upload your final files for publication. Because the schedule for publication is very tight, it is a condition of publication that you submit the revised version of your manuscript within 7 days. If you do not think you will be able to meet this date please let me know immediately.

To upload your manuscript, log into <http://mc.manuscriptcentral.com/prsb> and enter your Author Centre, where you will find your manuscript title listed under "Manuscripts with Decisions." Under "Actions," click on "Create a Revision." Your manuscript number has been appended to denote a revision.

You will be unable to make your revisions on the originally submitted version of the manuscript. Instead, upload a new version through your Author Centre.

1) A text file of the manuscript (doc, txt, rtf or tex), including the references, tables (including captions) and figure captions. Please remove any tracked changes from the text before submission. PDF files are not an accepted format for the "Main Document".

2) A separate electronic file of each figure (tiff, EPS or print-quality PDF preferred). The format should be produced directly from original creation package, or original software format. Please note that PowerPoint files are not accepted.

3) Electronic supplementary material: this should be contained in a separate file from the main text and the file name should contain the author's name and journal name, e.g. `authorname_procb_ESM_figures.pdf`

All supplementary materials accompanying an accepted article will be treated as in their final form. They will be published alongside the paper on the journal website and posted on the online figshare repository. Files on figshare will be made available approximately one week before the accompanying article so that the supplementary material can be attributed a unique DOI. Please see: <https://royalsociety.org/journals/authors/author-guidelines/>

4) Data-Sharing and data citation

It is a condition of publication that data supporting your paper are made available. Data should be made available either in the electronic supplementary material or through an appropriate repository. Details of how to access data should be included in your paper. Please see <https://royalsociety.org/journals/ethics-policies/data-sharing-mining/> for more details.

<http://datadryad.org/submit?journalID=RSPB&manu=RSPB-2020-2055> which will take you to your unique entry in the Dryad repository.

Once again, thank you for submitting your manuscript to Proceedings B and I look forward to receiving your final version. If you have any questions at all, please do not hesitate to get in touch.

Sincerely,

Dr Sasha Dall

Associate Editor

Board Member

Comments to Author:

Dear Dr Koenig,

One expert in the field has reviewed your manuscript. Considering the comments, I glad to recommend your work for publication on Proceeding of the Royal Society B.

Best wishes,

Roberto Feuda

Reviewer(s)' Comments to Author:

Referee: 2

Comments to the Author(s).

Dear authors,

Congratulations on your interesting manuscript. You made a big effort to address all points of criticism and to improve the quality of your manuscript.

Best,

Tim Wollesen

Sincerely,

Proceedings B

Decision letter (RSPB-2020-2055.R1)

28-Sep-2020

Dear Dr Koenig

I am pleased to inform you that your manuscript entitled "KLF/SP Transcription Factor Evolution in the Spiralia and the Implications for Cephalopod Visual System Novelty" has been accepted for publication in Proceedings B.

Open Access

Paper charges

All supplementary materials accompanying an accepted article will be treated as in their final form. They will be published alongside the paper on the journal website and posted on the online

figshare repository. Files on figshare will be made available approximately one week before the accompanying article so that the supplementary material can be attributed a unique DOI.

Sincerely,
Editor, Proceedings B
<mailto:proceedingsb@royalsociety.org>

Appendix A

This paper is about the evolution of KLF/SP family zinc finger proteins numbered in cephalopods. This family has been well studied in metazoans but with an underrepresentation of spiralia groups. The authors have listed the sequences available and established a phylogeny using two methods. They elaborate hypotheses on a higher diversification of these genes in spiralia. They characterized the genes in *Doryteuthis pealii* (Cephalopoda) and they focus on SP6-9 genes to study their expression during development. They highlight specifically the eye expression. In a third part they establish the chromosomal location of the three SP6-9 genes found in cephalopods and in mollusks and they link their observation to the diversity and the appearance of morphological novelties (with the eye as an example).

The study includes for the first time numerous spiralia sequences and this work is absolutely essential in the general context of of gene family evolution understanding. Actually, hypotheses are often based on vertebrates and sometimes ecdysozoan among protostomians. As a consequence of complex gene evolutionary stories, the role/function of genes are often also diversified and this paper contributes to the knowledge about the relationships between gene evolution and function. Cephalopods are now well known for their specificities, morphological innovation, and their origin, at the molecular level needs to be more explored.

Nevertheless the different results event if participating to the same general scheme of interests are too much disperse and not sufficiently discussed and/or robust to underpin the hypothesis of a link between the evolution of KLF/SP family in spiralia and the novelties (eye) in cephalopods.

In that sense, the results and discussion do not match to the title.

The authors expose interesting results but the link between the three (?) parts is not obvious, the congruence , the direction is not clear. The results are not really discussed.

Below are presented some comments on the three parts (identified as three parts)

Phylogeny and characterization of *D. pealii* genes

The concatenation of all spiralia sequences included in existing alignment of deuterostomia/ecdysozoa is a great work. Nevertheless, as said by the authors, these genes have a very conserved and characteristic domain, and a very diverged domain. Then the number of informative sites to establish the phylogeny is low either at the metazoan levels for the conserved domain or at the spiralia level for the diverged domain. The alignments have been automatically made by Clustal. Actually the variable part has been aligned randomly and information has been lost. It is sometimes necessary to check and align by eye. The analyses presented are not very clear about the number of informative sites that are taken into account to build the trees, the exact alignment taken into account to establish phylogeny (if it is the alignment presented in SEM, probably, they could be improved in some parts and some regions could be discarded).

Maybe as a consequence, the trees do not appear robust with few nodes well supported. Figure 1c is used for some explanation but not readable for the explanation given. Figure 1c indicate bootstrap % or posterior probability? It is not exactly the same range of confidence between bootstrap and posterior probability. The hypotheses built from PP below 0.95 are not generally considered as correct (whereas a node is considered as robust with a bootstrap value of 80%).

The description for each group of genes should be improved (in the introduction as well as in the tree). As the authors try to correlate the evolution, diversity, duplication (genomic events in general) to the function it would be necessary to give an overview of the different roles (or functions) of the different genes if possible.

Finally, for this part, it seems that the hypothesis of a diversification higher in spiralia than in vertebrates is difficult to see in the trees shown (fig 1c and SEM).

Roles and expression

The different roles are not clearly exposed for these genes to explain and/or compare and/or discuss the expression in cephalopods. The roles are very numerous and diverse but if the target is the visual system, maybe information could be brought to the genes of this family potentially involved in eye development, formation, and visual function in other groups.

The results of KLF/SP expression from a previous transcriptome is surprisingly exposed in the introduction whereas it is a result, not presented in the previous paper. In this last, Koenig et al, 2016, the database has been prepared and built and used for a specific purpose. As these results on KLF/SP were not presented in the 2016 paper, there must be maybe explained in this paper as a result. I suggest to present that these genes were found from these transcriptomes (belonging to one of the author). More information could be added regarding expression of these genes in other tissues in order to have a comparison level. Expression levels shown in fig.1b have an importance and constitute an argument only and only if the expression levels are lower in other tissues. Why the genes SP6-9 were chosen for the ISH take its origin in this result. But KLF should be also interesting. Obviously it is not possible to do expression for each gene but the choice of the specific family, SP6-9, KLF 5 (and not KLF15) could be more justified in that sense.

Expression in eye in figure 3: animals are not exactly at the same stage but the difference of expression between the two transcripts is clear. At stage 19, the eye begins to form by invagination. At this stage it is an eye placode and not a retina placode. Actually the retina begins to form and is clearly present when the vesicle begins to close after stage 21.

In SEM: expression of SP4-1 appears "ubiquitous" and closer to "noise" than to "true" expression. Without section it is difficult to be sure of the tissue location (in optic lobe in this case, sometimes expression surrounds, occupy conjunctive tissue and/or muscle). Conversely some expression of KLF5 are very clear, very interesting and not exploited. Expression is located in the cells of the yolk sac (and not in the yolk that is acellular). And other expression could be more discussed.

What is exactly novelties for the authors? There is no link between the evolution of the family and the eye except that there is an expression of these genes during development in eye and optic lobes (as many other genes probably). Maybe the focus on the eye here is not appropriate. By the results on expression in lentigenic tissue it could be justified but the lens, is not a novelty as the lens exists in Nautilus and numerous other organisms (such the Bivalvia...). Considering the diversity of expression in many tissues there is no reason to give more importance to the eye than to the other tissues. Moreover there were a lot of gene expression studied for the eye and these information are not truly included in the discussion section

Phylogeny, synteny and diversity

This part has been placed after the expressions data, probably to link the role/function and the genomic evolution by studying the synteny in cephalopods for which the genomes are available. One more it is a hard and essential work to approach the specificity of genomic organization in cephalopods and to find information about how the genomic diversity appear and/or could be responsible of appearance of role/function (neofunctionalisation). The discussion about the evolution/duplication of SP6-9 in cephalopods is a little bit confusing: if SP6 is duplicate and one is devoted to a new function, aka the lens building, what is its supposed role in Nautilus?

In the light of the results shown, the link established from the genomic organization to the morphological novelties such as the eye (visual function) would merit further investigation (or explanation).

In the text:

Line 18: "new genetic material": this must be precised.

For an interesting review about the link between innovation, genetics etc and the interest to study cephalopods, the paper of Ritschard et al, 2019 (in Bioessays) could be helpful for this purpose- I precise I am not a co-author.

Lines 20-22 face to line 25: are these two sentences in agreement?

Line 29: "little is known". Which other argument could the authors find to include other species than always the same models (ectodermata and vertebrates)?

Line 39: Interest to study this family is not clear

Line 47: "virtually unstudied" is not clear

Line 50: "eye and optic lobe tissue, an organ system": not exact, it can be changed

Line 52-65: these details on the eye development are not really justified with the beginning of the introduction and in the light of the results. See general comments.

M&M

It lacks a lot of information particularly about the number of AA used for the analysis and the differences obtained. It is indicated "sequences were truncated and concatenated" or "we used full length...when possible". Where is the information about the real length used available?

Line 117: a space is lacking

Line 119 and following: the classification is difficult to understand.

Results & Discussion

The subtitles and sections appear clearly separated but do not illustrate the results that could be more discussed

Line 154-156: these sentences will be better located in the introduction

Line 155: basal Metazoans? Which ones?

Line 182: "We hypothesis..sequences ": because what? Give more explanation for this reference (It should not be necessary to read the paper to understand your sentence).

Line 190: "We resolve two groups"; better to say: "The phylogeny shows two groups" (not very clear because the tree is small and the colors very similar).

There is a general problem with the results and the discussion around the SP1-5 and SP6-9 groups. The phylogeny does not show a clear distinction as the groups are not supported. But the most important is to understand clearly the evolution of this overall group and a phylogeny with alignment by-eye could maybe bring some informations. (cf Line 263).

Lines 195-196: maybe but not clear on the tree. In what it is "typical"? Gains and losses particularly in fishes are probably numerous (because of the genome duplications) and probably very "specific" also.

NB: KLF9-13 group is not resolved for spiralia because less constrained than other KLF/SP genes or because a lower number of phylogenetically significant AA?

Lines 199-201: this argument is not clear. (SP6-9c/a/b)

Line 225: "apparent later"?

Line 231 "anterior segment" is mostly used for ectodermata or metameric animals. Anterior part, side is more adapted especially in cephalopods where the orientation is either morphologic or physiologic.

Line 239: "limbs" is not used for cephalopods : arm is the right word (appendices eventually)

Line 244: what is indicated as anterior chamber organ is also the location of what is called white bodies with a role a little bit different and this information could be added (important when the role/function is analysed).

Line 257 SP6-9a is present in all cephalopod groups and not in Dp whereas it is lentigenic (but it was not found in embryo but maybe present in adult...). Moreover in the phylogeny, SP6-9a is closed to SP6-9-1 of gastropoda. The observation of the tree in SEM (fig S2) does not seem confirm the phylogenetic hypothesis elaborated.

Line 263: SP5/SP6-9 distinction is not clear (see above).

Line 264: they do not appear as sister groups in fig S2.

The section of discussion about the lensless nautilus is interesting but must be deeper discussed based on the results (see above)

Line 317: “connecting a cephalopod-specific gene duplication to a visual system neofunctionalization” is not really shown with the results

Line 320: “anterior segment” see comment on line 213

Details:

Conchifera and not Chonchifera

Line 296 and following: Clitellata and not Clitelata

Fig 5b: Pteriomorpha and not Pteriomorphia

Finally, considering the Ethics statement. Actually there is no ethics rule for cephalopods in United states but there is a European legislation that includes cephalopods in the welfare and all necessary to avoid pain and suffering in animals. The European legislation gives no indication about the cephalopods embryos. In that sense there is no obligation to mention any ethic statement.

Nevertheless 1) some American scientists follow the European legislation for their research and 2) the information above could be indicated in the text as the journal is a European one.

Appendix B

We thank the reviewers for their recognition of the importance of this work and their thoughtful comments and criticisms. We have performed additional phylogenetic analyses and revised the manuscript according to the reviewer's suggestions. Below, we have provided a point-by-point response to each comment.

Reviewer 1:

This paper is about the evolution of KLF/SP family zinc finger proteins numbered in cephalopods. This family has been well studied in metazoans but with an underrepresentation of spiralia groups. The authors have listed the sequences available and established a phylogeny using two methods. They elaborate hypotheses on a higher diversification of these genes in spiralia. They characterized the genes in *Doryteuthis pealii* (Cephalopoda) and they focus on SP6-9 genes to study their expression during development. They highlight specifically the eye expression. In a third part they establish the chromosomal location of the three SP6-9 genes found in cephalopods and in mollusks and they link their observation to the diversity and the appearance of morphological novelties (with the eye as an example).

The study includes for the first time numerous spiralia sequences and this work is absolutely essential in the general context of gene family evolution understanding. Actually, hypotheses are often based on vertebrates and sometimes ecdysozoan among protostomians. As a consequence of complex gene evolutionary stories, the role/function of genes are often also diversified and this paper contributes to the knowledge about the relationships between gene evolution and function. Cephalopods are now well known for their specificities, morphological innovation, and their origin, at the molecular level needs to be more explored.

We thank the reviewer for recognizing the importance of this work.

Nevertheless the different results event if participating to the same general scheme of interests are too much disperse and not sufficiently discussed and/or robust to underpin the hypothesis of a link between the evolution of KLF/SP family in spiralia and the novelties (eye) in cephalopods.

We thank the reviewer for the comment. We have added text to better highlight the rationale for our focus on SP6-9 as it relates to cephalopod novelty and the connections between our experiments throughout the manuscript.

In that sense, the results and discussion do not match to the title.

We hope our expanded text addresses the reviewer concerns and supports the title.

The authors expose interesting results but the link between the three (?) parts is not obvious, the congruence, the direction is not clear. The results are not really discussed.

With sensitivity to length constraints, we have added more discussion of the results into the manuscript (See Discussion) and expanded on our rationale connecting the three parts.

Below are presented some comments on the three parts (identified as three parts)

Phylogeny and characterization of D.pealii genes

The concatenation of all spiralia sequences included in existing alignment of deuterostomia/ecdysozoa is a great work. Nevertheless, as said by the authors, these genes have a very conserved and characteristic domain, and a very diverged domain. Then the number of informative sites to establish the phylogeny is low either at the metazoan levels for the conserved domain or at the spiralia level for the diverged domain. The alignments have been automatically made by Clustal. Actually the variable part has been aligned randomly and information has been lost. It is sometimes necessary to check and align by eye. The analyses presented are not very clear about the number of informative sites that are taken into account to build the trees, the exact alignment taken into account to establish phylogeny (if it is the alignment presented in SEM, probably, they could be improved in some parts and some regions could be discarded).

We appreciate that the reviewer has highlighted this issue, because it is one we struggled with while phylogenetically characterizing these sequences. We have tried multiple methods to generate each tree in this work but these efforts did not improve our trees. The trees presented are the best representatives. We therefore narrowed our phylogenetic analyses throughout the manuscript to make more confident conclusions.

For the metazoan KLF/SP family tree (Figure 1C), we only used the conserved motifs and C2H2 domains identified by Presnell et al. 2014 in our alignments [1]. As the reviewer has suggested, for this large alignment, we already eliminated uninformative sites by using only the conserved domains (Data S1).

The Spiralian SP6-9 trees (supplementary figure S3) are a result of multiple iterations of full-length and trimmed sequence alignments (using TrimAl and Gblocks), using ClustalW, Muscle, MAFFT, and hand curation during the alignment and generating both ML and Bayesian trees. We found that none of these methods improved our branch support. For the trees included in the manuscript, the final alignments were not trimmed, however, the phylogenetic focus in the Spiralia made a non-random alignment outside of well characterized domains (Data S4). Furthermore, we do not feel confident in manual changes

in this case, as homologous regions of each sequence are not clear outside the binding domain. We therefore left the alignment to the heuristic algorithm.

Our cephalopod-specific SP6-9 tree (Figure 4a) has significant branch support and we feel that the alignment method has done a good job for these more closely-related sequences.

Due to the well-described issues with this gene family, it is no surprise our trees are not well-resolved, but the methods development required for better phylogenetic inference is beyond the scope of this manuscript.

Maybe as a consequence, the trees do not appear robust with few nodes well supported. Figure 1c is used for some explanation but not readable for the explanation given. Figure 1c indicate bootstrap % or posterior probability? It is not exactly the same range of confidence between bootstrap and posterior probability. The hypotheses built from PP below 0.95 are not generally considered as correct (whereas a node is considered as robust with a bootstrap value of 80%).

We agree with the reviewer that many of the deeper nodes are not well-supported in this tree. This is a consequence of the sequence architecture of this gene family as noted above, and in previous literature. These support values are posterior probabilities, as mentioned in the figure legend, but we have also added this information in the results section. Our tree does not change the state of SP and KLF groups as established in Presnell et al. 2015 and Schaeper et al., 2010. Increased depth and breadth of sequencing of problematic groups, particularly KLF10/11 and KLF9/13, could improve these trees, but is not our focus in this paper. Difficulty assessing this gene family with currently available sequences led us to narrow our focus to SP5-9 genes, in an effort to generate a more robust inference.

The description for each group of genes should be improved (in the introduction as well as in the tree). As the authors try to correlate the evolution, diversity, duplication (genomic events in general) to the function it would be necessary to give an overview of the different roles (or functions) of the different genes if possible.

We appreciate the reviewer's comment, and share their enthusiasm for describing the gene family. We limited our descriptions primarily due to space constraints and the overwhelming diversity of functions of this gene family. At the reviewer's request we have included more background about the range of KLF/SP functions, with a focus on SP6-9.

Finally, for this part, it seems that the hypothesis of a diversification higher in spiralia than in vertebrates is difficult to see in the trees shown (fig 1c and SEM).

We apologize for our lack of clarity. We agree with the reviewer that Spiralian KLF/SP diversity is not greater than that of vertebrates. Our hypothesis is more specifically that

KLF/SP genes in the spiral lineage have expanded since the bilaterian ancestor and is summarized in figure 5. This expansion is not found in ecdysozoans and basal deuterostomes and could not be assumed before this current analysis. We have edited the text to clarify.

Roles and expression

The different roles are not clearly exposed for these genes to explain and/or compare/and /or discuss the expression in cephalopods. The roles are very numerous and diverse but if the target is the visual system, maybe informations could be bring to the genes of this family potentially involved in eye development, formation, and visual function in other groups.

We've added additional background for these genes in the introduction.

The results of KLF/SP expression from a previous transcriptomes is surprisingly exposed in the introduction whereas it is a result, not presented in the previous paper. In this last, Koenig et al, 2016, the database has been prepared and built and used for a specific purpose. As these results on KLF/SP were not presented in the 2016 paper, there must be maybe be explained in this paper as a result. I suggest to present that these genes were found from these transcriptomes (belonging to one of the author).

We've moved this description of the expression levels to the results.

More information could be added regard to expression of these genes in other tissues in order to have a comparison level. Expressions levels shown in fig.1b have an importance and constitute an argument only and only if the expression levels are lower in other tissues.

This RNA-seq experiment and methods has been previously published from a time course experiment of eye and optic lobe tissue (Koenig et al. 2016). For a developmental time course experiment, relative expression is not statistically compared to other tissues, it is internally tested. Determining significance within time course data is a non-trivial problem and is still not solved. Raw reads are normalized and expression is reported over time relative to other genes and developmental stages. Our focus on the SP/KLF genes is justified because they are enriched in visual system development relative to other genes as shown by our heatmap. A gene can be highly expressed in other tissues and still be of considerable interest to development and innovation in the visual system. We have clarified our language to illuminate this difference.

Why the genes SP6-9 were chosen for the ISH take its origin in this result. But KLF should be also interesting. Obviously it is not possible to do expression for each gene but the choice of the specific family, SP6-9, KLF 5 (and not KLF15) could be more justified in that sense.

Based on the high visual system expression levels and the opportunity to focus on a gene duplication, we chose to further investigate SP6-9a and b paralogs in the squid. This rationale is now expanded in the text. We did clone and perform *in situ* hybridization for KLF15 but were unsuccessful (Supplementary Table S1).

Expression in eye in figure 3: animals are not exactly at the same stage but the difference of expression between the two transcripts is clear. At stage 19, the eye begins to form by invagination. At this stage it is an eye placode and not a retina placode. Actually the retina begins to form and is clearly present when the vesicle begins to close after stage 21.

We apologize to the reviewer for our lack of clarity in differentiating the eye placode from the retina placode. Previous studies using Dil lineage-tracing have shown that the retina is derived from placodal tissue and the anterior segment is derived from the internalizing lip cells at stage 18-20, after which the vesicle has closed [2]. We have added additional language to clarify the difference between the placode and the lip tissue during development and changed the annotations in Figure 3 to better show this process.

In SEM: expression of SP4-1 appears “ubiquitous” and closer to “noise” than to “true” expression. Without section it is difficult to be sure of the tissue location (in optic lobe in this case, sometimes expression surrounds, occupy conjunctive tissue and/or muscle).

We agree with the reviewer that ubiquitous expression can be difficult to distinguish from noise in *in situ* data. We have done this experiment multiple times, with sense probes, and this expression is qualitatively different from background and repeatable over developmental stages. “Noise” relative to specific expression is not restricted to developing tissues and is found ubiquitous in the yolk as well, unlike in the SP1-4 probe. Finally, ubiquitous expression was expected as previously published expression of SP1-4 is ubiquitous in multiple arthropod and vertebrate species during development [3,4]

Conversely some expression of KLF5 are very clear, very interesting and not exploited. Expression is located in the cells of the yolk sac (and not in the yolk that is acellular). And other expression could be more discussed.

We agree that this expression is interesting, and why we chose to include it, but we regret that space constraints prevent us from expanding our discussion here.

What is exactly novelties for the authors? There is no link between the evolution of the family and the eye except that there is an expression of these genes during development in eye and optic lobes (as many other genes probably). Maybe the focus on the eye here is not appropriate. By the results on expression in lentigenic tissue it could be justified but the lens, is not a novelty as the lens exists in Nautilus and numerous other organisms (such the Bivalvia...).

Considering the diversity of expression in many tissues there is no reason to give more importance to the eye than to the other tissues. Moreover there were a lot of gene expression studied for the eye and these information are not truly included in the discussion section

We respectfully disagree with the reviewer that the coleoid cephalopod eye is not an appropriate focus for the study of novel traits.

The literature shows that the Nautilus does not have a lens and its visual organ functions as a pinhole-type eye [5–7] . The lens is a novelty to coleoid cephalopods and dramatically improves visual acuity of these taxa [6] The lens also exists in bivalves but this is widely accepted to be an incidence of convergence [5,7].

SP6-9a and b are similar in sequence and expression pattern with the exception of the developing lens. One of our goals was to understand if C2H2 zinc finger transcription factor duplications contributed to visual system novelties. The heatmap supports our *in situ* investigation of these paralogs and our expression patterns support a new function for one of these paralogs in the development of a visual system novelty. This finding fits in a historical context highlighting visual system innovations as important examples in the study of the evolution of novelty and complexity [8–12]. We have expanded our background to include a better justification for our choice to focus on the eye and SP6-9 [6].

Phylogeny, synteny and diversity

This part has been placed after the expressions data, probably to link the role/function and the genomic evolution by studying the synteny in cephalopods for which the genomes are available. One more it is a hard and essential work to approach the specificity of genomic organization in cephalopods and to find informations about how the genomic diversity appear and/or could be responsible of appearance of role/function (neofunctionalisation). The discussion about the evolution/duplication of SP6-9 in cephalopods is a little bit confusing: if SP6 is duplicate and one is devoted to a new function, aka the lens building, what is its supposed role in Nautilus?

We apologize to the reviewer that this was not clear in the text. As mentioned above, Nautilus do not have a lens, however the function of this gene in Nautilus is an interesting question. In the text we mention that short branch lengths within the SP6-9a clade suggest that sequence changes are not altering the protein function from Nautilus to squid. Instead, *cis*-regulation has likely altered the domain of expression to include lens-making tissue in coleoid cephalopods, leading to a new role.

In the light of the results shown, the link established from the genomic organization to the morphological novelties such as the eye (visual function) would merit further investigation (or explanation).

We apologize to the reviewer for our lack of clarity. We do not intend to make the argument in the text that genome organization relates to morphological novelties like the eye. Due to inconclusive phylogenetic support, we wanted additional evidence to explore SP6-9 evolutionary history in cephalopods. Recent paralogs are often found in tandem, and mechanisms of duplication can be elucidated from exon/intron structure, therefore, we assessed genomic architecture in cephalopods to gain greater insight. We have rewritten this rationale in the main text to improve clarity.

In the text:

Line 18: “new genetic material”: this must be precised.

For an interesting review about the link between innovation, genetics etc and the interest to study cephalopods, the paper of Ritschard et al, 2019 (in Bioessays) could be helpful for this propose-I precise I am not a co-author.

We want to thank the reviewer for the comment and for the direction to this interesting review. We have expanded our definition of “new genetic material” in the following sentence as informed by the helpful citation provided.

Lines 20-22 face to line 25: are these two sentences in agreement?

We have clarified the language.

Line 29: “little is known”. Which other argument could the authors find to include other species than always the same models (ecdysozoa and vertebrates)?

We’ve changed this line to compare cephalopod novelty to better studied animal systems.

Line 39: Interest to study this family is not clear

C2H2 zinc finger transcription factors are an understudied group as a whole. In light of the expansion of this family in cephalopods and its potential to contribute to lineage specific novelties, we chose one of the best studied sub-families as a starting point. We have made this connection clearer in the text.

Line 47: “virtually unstudied” is not clear

We have replaced with “poorly understood”

Line 50: “eye and optic lobe tissue, an organ system”: not exact, it can be changed

Addressed.

Line 52-65: these details on the eye development are not really justified with the beginning of the introduction and in the light of the results. See general comments.

We’ve strengthened the rationale for highlighting the novelty of the cephalopod lens and camera-type eye. With a focus on how transcription factor duplications could contribute to the formation of novel structures in the eye, we believe an introduction to eye development is necessary background for the reader. .

M&M

It lacks a lot of information particularly about the number of AA used for the analysis and the differences obtained. It is indicated “sequences were truncated and concatenated” or “we used full length...when possible”. Where is the information about the real length used available?

We have provided all the alignment files in universal fasta format where read length is easily assessed (Supplemental Data Files 1, 4, 7, 9, and 11).

Line 117: a space is lacking

Addressed.

Line 119 and following: the classification is difficult to understand.

We have clarified the language.

Results & Discussion

The subtitles and sections appear clearly separated but do not illustrate the results that could be more discussed

We hope that our expanded discussion addresses the reviewer’s concern.

Line 154-156: these sentences will be better located in the introduction

Our introductory language gives a broader perspective on our question and we hope our expanded text clarifies our motivations to the reviewer. This tree is only one part of the story and therefore is discussed later in the paper.

Line 155: basal Metazoans? Which ones?

We've included a reference to a supplementary table that lists all of the species included in our analysis (Table S2).

Line 182: "We hypothesis..sequences ": because what? Give more explanation for this reference (It should not be necessary to read the paper to understand your sentence).

Addressed.

Line 190: "We resolve two groups"; better to say: "The phylogeny shows two groups" (not very clear because the tree is small and the colors very similar).

This text change and color scheme has been addressed.

There is a general problem with the results and the discussion around the SP1-5 and SP6-9 groups. The phylogeny does not show a clear distinction as the groups are not supported. But the most important is to understand clearly the evolution of this overall group and a phylogeny with alignment by-eye could maybe bring some informations. (cf Line 263).

We appreciate the reviewer's concern and have softened some of the language regarding our confidence in these two groups. As mentioned above, our KLF/SP tree is generated from only the conserved domains, so all sequences are well-aligned (Data S1). However, the reduced number of residues in the alignment may be impacting our phylogenetic inference. To address this, the remaining paper does significant work to show the phylogeny of the SP5-9 group with taxonomically focused analyses. These more narrow analyses all show SP5 highly supported as a separate group from SP6-9.

Lines 195-196: maybe but not clear on the tree. In what it is "typical"? Gains and losses particularly in fishes are probably numerous (because of the genome duplications) and probably very "specific" also.

We agree with the reviewer that the gains and losses in other lineages such as fishes are likely numerous. However, we do not intend to make a statement on whether this pattern is typical or atypical in evolution, but rather that these occurrences happened specifically in the spiralian lineage. Basal deuterostomes and the majority of ecdysozoans sampled do not have the number of duplications observed in either spiralian or vertebrate species. Many of these spiralian duplications also form clades to the exclusion of vertebrates. The most parsimonious

conclusion from these data is that the KLF/SP gene family has independently undergone a series of spiralian-specific gains and losses.

NB: KLF9-13 group is not resolved for spiralia because less constrained than other KLF/SP genes or because a lower number of phylogenetically significant AA?

In our trimmed alignment of conserved domains, some SP/KLF sequences have more phylogenetically significant amino acids in the alignment than others, however this does not correlate with support in the tree. (Data S1). Other groups have also had difficulty resolving the KLF9/13 and KLF10/11 groups and these are paraphyletic in Presnell et al. 2015. It may be that reduced evolutionary constraint is at play but analyses beyond the scope of this manuscript would be required to assess this.

Lines 199-201: this argument is not clear. (SP6-9c/a/b)

We have expanded the text to clarify this decision.

Line 225: “apparent later”?

We have clarified our language.

Line 231 “anterior segment” is mostly used for ecdysozoa or metameric animals. Anterior part, side is more adapted especially in cephalopods where the orientation is either morphologic or physiologic.

We appreciate the reviewer’s preferences but respectfully disagree. In the field of visual system development, the anterior of a single-chambered eye, including the lens, cornea, iris and pupil, is referred to as the anterior segment. We therefore have chosen this vocabulary as the most consistent and clear word choice. We have further clarified our language with the addition of a anatomical cartoon of the tissue (Supplemental Figure S2)

Line 239: “limbs” is not used for cephalopods : arm is the right word (appendices eventually)

We have chosen to use the embryonic nomenclature as outlined in Nodl, et al. 2015, Nodl et al. 2016, and Tarzona et al. 2019. [13–15]

Line 244: what is indicated as anterior chamber organ is also the location of what is called white bodies with a role a little bit different and this information could be added (important when the role/function is analysed).

We agree that some descriptions of embryonic development identify the region we have indicated as the anterior chamber organ as the location of the white bodies, and that the

white bodies have an alternate function. Unfortunately, it is our assessment of the literature that the anterior chamber organ is the only tissue whose development has been well enough described to be identified [16–18].

Line 257 SP6-9a is present in all cephalopod groups and not in Dp whereas it is lentigenic (but it was not found in embryo but maybe present in adult...).

We apologize to the reviewer for our lack of clarity, it is SP6-9c that is found in other cephalopods but not in *Doryteuthis*. SP6-9a is found in *Doryteuthis* and is expressed in the lentigenic cells, while SP6-9b is found in *Doryteuthis* but not expressed in the lentigenic cells. It is likely that SP6-9c exists in *Doryteuthis* but its absence from our embryo-specific transcriptome, suggests it is not expressed at these stages.

We agree with the reviewer that our current data does not rule out a role for SP6-9b or c in the adult squid lens. However, this possibility does not exclude the neofunctionalization of SP6-9a in development.

Moreover in the phylogeny, SP6-9a is closed to SP6-9-1 of gastropoda. The observation of the tree in SEM (fig S2) does not seem confirm the phylogenetic hypothesis elaborated.

(Now Figure S3) We apologize to the reviewer if this was not clear, but although gastropoda SP6-9-1 and cephalopod SP6-9a are physically close in our trees (fig S3), there is no support for this relationship. Nodes without circles denote lower than 50% support in both trees and we have added this annotation more explicitly in the legend.

Line 263: SP5/SP6-9 distinction is not clear (see above).

We agree that our previous trees did not resolve SP6-9 from SP5 and we made the cephalopod tree to address this problem. In this cephalopod tree (and our mollusc and annelid specific trees), our SP6-9 node has a 95% posterior probability, strongly supporting the existence of these two groups, SP5 and SP6-9. (Figure 4a, Supplementary Figures S7, S8)

Line 264: they do not appear as sister groups in fig S2.

We agree that in the Spiralian tree of the SP6-9 group, SP6-9a and SP6-9b do not appear sister. Rather, both Bayesian and Maximum likelihood trees (Figure S3) are inconclusive in their relationships to one another. This is why we generated a specific cephalopod tree shown in Figure 4A. In Figure 4A, we see a 74% pp support for SP6-9a and SP6-9b as sister to SP6-9c.

The section of discussion about the lensless nautilus is interesting but must be deeper discussed based on the results (see above)

It is our hypothesis that SP6-9 has similar biochemical function in Nautilus and *D. pealeii*, but that the domain of expression allows SP6-9a novel DNA-binding in a distinct chromatin environment leading to a new function in lens development arising in coleoids.

Line 317: “connecting a cephalopod-specific gene duplication to a visual system neofunctionalization” is not really shown with the results

We hope that our expanded discussion helps clarify our case. Neofunctionalization is defined as the duplication of a gene and the divergence of a paralog in sequence and/or expression pattern [19–21]. In this study, we identify a cephalopod specific transcription factor duplication as supported by expression pattern, synteny, gene architecture, as well as phylogenetics. Coinciding with this gene duplication is the emergence of a paralog-specific expression domain in the lens, a coleoid cephalopod novelty. This new expression domain suggests a new regulatory function, with novel targets, in the cephalopod anterior segment. It is our assessment that these data support a duplication and neofunctionalization trajectory.

Line 320: “anterior segment” see comment on line 213

See above

Details:

Conchifera and not Chonchifera

Addressed.

Line 296 and following: Clitellata and not Clitelata

Addressed.

Fig 5b: Pteriomorpha and not Pteriomorphia

We have derived our nomenclature from a recent phylogenetic assessment of molluscs, Lemer et al. 2019, as well as the World Register of Marine Species, which state Pteriomorphia [22].

Finally, considering the Ethics statement. Actually there is no ethics rule for cephalopods in United states but there is a European legislation that includes cephalopods in the welfare and

all necessary to avoid pain and suffering in animals. The European legislation gives no indication about the cephalopods embryos. In that sense there is no obligation to mention any ethic statement. Nevertheless 1) some American scientists follow the European legislation for their research and 2) the information above could be indicated in the text as the journal is a European one.

As mentioned by the reviewer, there are no ethics rules in Europe or the United States regarding the treatment of cephalopod embryos. However, the Koenig lab does follow European guidelines in our treatment of cephalopods to avoid pain and suffering in animals. We have included a statement to this effect.

Reviewer 2:

Comments to the Author(s)

McCulloch and Koenig study the expression, orthology, and synteny of members of the KLF/SP gene family in the Spiralia with special focus on the cephalopod *Doryteuthis pealeii*. Their work follows up on aspects of a previous study which revealed an enriched expression of KLF/SP genes in the developing eyes of squid (König et al. 2016). In the present manuscript the authors mined the transcriptome of cephalopods as well as other spiralian for members of this gene family. The authors performed in situ hybridization experiments on several members of this family in squid developmental stages. Using published genomes and transcriptomes they conclude that the individual spiralian clades have divergent numbers of SP6-9 paralogs which show synteny in cephalopods but not the other spiralian. Among the squid paralogs, SP6-9a is expressed in the squid lens in contrast to SP6-9b leading the authors to infer that gene duplication led to neofunctionalization of this paralog.

The study is well-conceived with convincing gene expression patterns and sound phylogenetic analyses. It contributes to our understanding on the evolution of coleoid cephalopod complex eyes and poses many more questions such as how these genes might be functional in *Nautilus* or the eyeless scaphopods.

We thank the reviewer for their positive assessment of this study. We appreciate the recognition of the importance of this work and how it opens the door to new questions in Spiralia biology and evolution.

I have few points of criticism that should be easily addressed:

Major issues:

a.) Some clades are definitively underrepresented in the ancestral state reconstruction shown in Fig.5b, c. In particular, basal-branching taxa such as the aculiferan Caudofoveata and Solenogastres are not included although transcriptomes do exist. There should also be publicly available transcriptomes of basally branching annelids such as *Owenia* and *Chaetopterus* (check work by C. Bleidorn and C. Helm for phylogeny and possibly transcriptomes). It is important to include these taxa since they may change the outcome of the analysis.

We thank the reviewer for identifying data missing in our tree. We have now assembled the transcriptomic data from Bleidorn, Helm, et al., and thoroughly searched our assemblies for SP6-9 homologs. The sequences we used as well as transcriptomes we searched but found no SP6-9 and all references are now listed in Supplementary Tables S2 and S4. We have identified and added new basal annelid and aculiferan sequences in our SP6-9 dataset, and reconstructed two new annelid and mollusc Bayesian trees. With the new data we have improved phylogenetic resolution for character mapping SP6-9 orthologs in both molluscs and annelids. This additional data provides more evidence for a single ortholog at the base of both Annelida and Mollusca.

The authors state “branches for which we have data” (l. 139). I have to admit that it is easy to miss out on new published transcriptomes, but the authors should definitively include the above-mentioned ones. If no orthologs are found, then this should be stated in the figure legend.

We previously searched multiple annelid, solenogastre, and caudofoveata transcriptomic datasets, but could not identify an SP6-9 ortholog, or only tiny fragments. We recognize that we should have more clearly stated which transcriptomes were searched and what we found. We have now included a table of all the transcriptomes that we searched, number of SP6-9 orthologs found, and their references (Supplementary Table S4).

b.) Why do the authors argue for one SP6-9 ortholog instead of two paralogs in the last common molluscan ancestor? In my opinion, an evolutionary scenario favoring two paralogs would be more parsimonious since less changes would be necessary? Maybe I overread some arguments in the manuscript?

As suggested by the reviewer, it is possible that the last common molluscan ancestor had two paralogs, especially because our dataset surveys may be incomplete. However, with the inclusion of the new data suggested by the reviewer we now have more confidence in our hypothesis. In all aculiferan transcriptomes searched, we never found more than one SP6-9 homolog. Therefore, a hypothesis placing the duplication at the base of molluscs would

require a loss in the Aculifera lineage and would be less parsimonious. We thank the reviewer for encouraging us to include these additional datasets.

c.) The authors barely cite references and their own beautiful figures in their manuscript (see below).

We want to thank the reviewer for this comment. We have added all the highlighted citations and references to figures.

d.) The figure (legends) should be self-explaining: Often gene expression domains are not labeled properly in the figures and labels are not explained in the legends (see below).

We have expanded our annotation in our gene expression figures and the detail in the legends.

Minor issues:

l. 28: single-chambered eyes

Addressed.

l. 37: which other species? provide references.

Addressed.

l. 45: I reckon it might be important to go a little more into detail about the function of these genes in non-spiralian taxa (only few examples).

Addressed.

l. 49: “developing eye and optic lobe tissue”: replace “tissue” with “tissues”. Here it would be important to provide an explanation for laypersons who are not familiar with the cephalopod nervous system (“optic lobe”).

Addressed.

l. 50: same here, please explain “Coleoid”. It should also be written lowercase. A reference for “significant innovations...” would be useful.

Addressed.

l. 53: here I would write coleoid cephalopods, since - as the authors stated - nautilus have different eyes (“pinhole-type eyes)

Addressed.

l. 56. Here references are missing – check throughout manuscript

Addressed.

l. 64. I would recommend including a sketch drawing of the eyes including all those layers, lens etc.

Addressed. See Supplemental figure S2

l. 70: This “cephalopod duplications” sounds awkward, better “we identify duplications of the KLF/SP gene families in cephalopods...”

Addressed.

l. 77: replace “these studies” with “our study”

Addressed.

l. 84: “kept in flowing seawater” – same water temperature?

At the MBL, embryos were kept at the local ocean water temperature which is approximately 20 degrees C with some variability. This variability in husbandry is common in wild caught species and makes morphological staging required for embryonic studies.

l. 89: “PCR products were cloned”, but size-fractionated before, right?

We may be misunderstanding the reviewer, but if they are asking whether we ran our PCR product on a gel to confirm size, we can affirm that we did this step. However, we assessed that size selection was not necessary and we confirmed our cloned insert using Sanger sequencing.

l. 117: “dentifiedSP6-9” space missing

Addressed.

l. 133 “Annelids which lack a chromosome-level assembly.”: Reference is missing and Annelids should be written in lower-case.

Addressed.

l. 137: “current hypotheses of relationships”: add “phylogenetic”

Addressed.

l. 138: annelids, spiralian, molluscs, BUT Spiralia, Annelida, Mollusca... Please correct throughout manuscript

Addressed.

l. 156: Reference missing

Addressed.

I. 158: “To produce the alignment...” this sentence should be part of the M&M
We have included this information in the M&M but we reference it here as well because it is important to understanding our phylogenetic results and our interpretation.

I. 180: metazoan, not Metazoan
Addressed.

I. 181: ref missing
Addressed.

I. 186: cite Figure
Addressed.

I. 186: I would rephrase the sentence “D. melanogaster does not have a 187 member of KLF10/11, but we do find Tribolium, Limulus, Daphnia, and multiple spiralian 188 sequences.” to “D. melanogaster does not possess a KLF10/11 gene but orthologs are found in ...”
Addressed.

I. 190: cite Figure
Addressed.

I. 214: I would refer to polyplacophoran mollusks since you did not introduce polyplacophorans before (not everyone in the audience is familiar with the individual molluscan clades). Same with scaphopods.
Addressed.

I. 222: sounds like Arnold did the in situ, please refer also to your figure ;)
Addressed.

I. 225: how do you define early development? Better mention stages, at least in parenthesis.
Addressed.

I. 225: “Arm and mantle expression is apparent later” – that contradicts the previous sentence (you already mentioned “arm” development)
We have clarified this language.

Fig. S3: expressed in cells covering the yolk sac. Inside the yolk sac as well?

(Now figure S4) The yolk itself is anuclear. We only see expression on the outside of the sac.

I. 229: Why do the authors discuss SP5 expression here again? I would reorganize this paragraph accordingly.

We have clarified our language as this is the first time we discuss SP5.

I. 237: Citation of Fig. is misleading since you also cite stages stage 24 and 27: Better cite "Fig. 2b-c, g-h..."

Addressed.

I. 237: better: SP6-9...expression in retina may be

Addressed.

I. 240: reference missing

Addressed.

I. 240-242: A role in what? I would combine both sentences and write "...limp outgrowth and our data suggest that both paralogs of SP6-9 may play a similar role"

We have clarified this language.

I. 246: "Despite their similar expression, these paralogs do show differences." In what? Better phrase differently such as: "despite shared expression domains, these paralogs are also expressed in different regions.

We have clarified this language.

I. 248: "The most dramatic", I would replace it with "Major differences"

Addressed.

I. 247: Why do you assume that these expression domains correspond to the cerebral and palliovisceral ganglia, respectively? I would expect the cerebral ganglia to be located more anteriorly and not that dorsal during this early dev. stage. Do you have any neural markers that would support your notion?

We appreciate the reviewer's comment because the development of these brain regions have not been well studied, especially in early developmental stages and in this specific species.

We base this identification on embryonic anatomy described in *Sepia officinalis* [23]. In addition, expression of *Six3/6* in the squid *D. pealeii*, a common marker for the most anterior embryonic neural tissue, includes the region we have identified as the cerebral ganglion [2].

However, despite this previous work, these cells could contribute to other brain regions as an in-depth lineage has not been generated. We have softened the language.

Also, please label both expression domains with differently colored arrows/ other labels.

Addressed.

I. 273: ...is correlated with coleoid lens evolution.

Addressed.

I. 281: in all cephalopod genomes investigated so far.

Addressed.

I.290: "This hypothesis suggests", better "we hypothesize.." or "we infer"

Addressed.

I. 291: This could be misunderstood. Better: "...Annelida, Mollusca, and the lineage leading to Phoronida and Bryozoa".

Addressed.

Please also change "Chonchifera" to "Conchifera"

Addressed.

I. 296: Clitellata

Addressed.

I. 300: Why did the authors not include basally-branching annelids such as Owenia and Chaetopterus (see above)?

As discussed above, we have now assembled transcriptomes and generated an annelid-specific tree including these basally-branching annelids (Supplemental figure S7). We find only a single SP6-9 ortholog in basal annelids, if we find any sequences in their respective transcriptomes. This suggests that at the base of annelids was a single SP6-9 ortholog, that has undergone duplications within Sedentaria.

Figure legend 5: It is important to cite the studies of the phylogenetic trees on which the characters are plotted.

Addressed.

I. 302: Please add references with respect to the phyl. studies (see above) e.g. (see Kocot et al. 2011, Smith et al. 2011?).

Addressed.

I.302: I am not entirely happy with the molluscan tree and the annelid tree since basally-branching taxa are underrepresented (see above). There are other aculiferan transcriptomes than the polyplacophoran one, e.g. of solenogastres and caudofoveates (De Oliveira AL, Wollesen T, Kristof A, Scherholz M, Redl E, Todt C, Bleidorn C, Wanninger A. 2016. Comparative transcriptomics enlarges the toolkit of known developmental genes in mollusks. BMC Genomics 17: 905 (23 pages). DOI: 10.1186/s12864-016-3080-9).

We hope our assembly of additional transcriptomic resources, the generation of additional trees and the discussion above addresses these concerns.

An increased number of paralogs (e.g. 2) among the Aculifera would change the phylogenetic scenario. Then polyplacophorans would have probably lost a paralog and a duplication event in the conchiferan stem lineage would have been unlikely. A similar scenario could also be possible for the annelid tree.

After addressing the reviewer's useful comments above, we now show that all Aculifera and basally branching annelid lineages have only a single paralog, if any SP6-9 transcript is found at all. This strengthens our hypothesis that a duplication occurred at the base of Conchifera, and independently multiple duplication events occurred in a subset of Sedentaria. We thank the reviewer for pointing us to these data and strengthening the conclusions of the paper.

I.303: Conchifera (see also Fig. 5)

Addressed.

I. 303: Cite Figure

Addressed.

I. 320: "in the anterior segment." Specify of what?

We have clarified this language.

Discussion

The authors state in the introduction that KLF/SP proteins have "diverse developmental, cellular and homeostatic" functions. Are these genes also implicated in eye development in other organisms? If yes, it would be worth discussing.

We have expanded our discussion of the role of KLF/SP genes in the introduction and discussion.

Figures:

Although the authors provide a sketch drawing depicting Dp developmental stages, it is necessary to label all structures in the micrographs that express a given gene (e.g. gills and arms in Figure S3), otherwise it is too cumbersome when reading the text body and checking the Figures.

Addressed (Now Figure S4).

Fig. 4a: Explain branch support (filled grey, black, white circles)

Addressed.

Fig. 5:

- 5c: What's the difference between a black and a grey "X" (gene loss). Should be indicated in the legend.

This was an unintentional difference and has been corrected.

- 5a: How many paralogs did the last common spiralian ancestor have?

Addressed.

1. Presnell JS, Schnitzler CE, Browne WE. 2015 KLF/SP Transcription Factor Family Evolution: Expansion, Diversification, and Innovation in Eukaryotes. *Genome Biol. Evol.* **7**, 2289–2309. (doi:10.1093/gbe/evv141)
2. Koenig KM, Sun P, Meyer E, Gross JM. 2016 Eye development and photoreceptor differentiation in the cephalopod *Doryteuthis pealeii*. *Development* **143**, 3168–3181. (doi:10.1242/dev.134254)
3. Schaeper ND, Prpic N-M, Wimmer EA. 2010 A clustered set of three Sp-family genes is ancestral in the Metazoa: evidence from sequence analysis, protein domain structure, developmental expression patterns and chromosomal location. *BMC Evol. Biol.* **10**, 88. (doi:10.1186/1471-2148-10-88)
4. Xie J, Yin H, Nichols TD, Yoder JA, Horowitz JM. 2010 Sp2 is a maternally inherited transcription factor required for embryonic development. *J. Biol. Chem.* **285**, 4153–4164. (doi:10.1074/jbc.M109.078881)
5. Jonasova K, Kozmik Z. 2008 Eye evolution: Lens and cornea as an upgrade of animal visual system. *Seminars in Cell & Developmental Biology.* **19**, 71–81.

(doi:10.1016/j.semcdb.2007.10.005)

6. Muntz WRA. 1987 Visual Behavior and Visual Sensitivity of *Nautilus pompilius*. *Topics in Geobiology*. , 231–244. (doi:10.1007/978-1-4899-5040-6_15)
7. Land MF. 2012 The evolution of lenses. *Ophthalmic and Physiological Optics*. **32**, 449–460. (doi:10.1111/j.1475-1313.2012.00941.x)
8. Darwin C. 1871 On the origin of species. (doi:10.5962/bhl.title.28875)
9. Shubin N, Tabin C, Carroll S. 2009 Deep homology and the origins of evolutionary novelty. *Nature* **457**, 818–823. (doi:10.1038/nature07891)
10. Salvini-Plawen L v., v. Salvini-Plawen L, Mayr E. 1977 On the Evolution of Photoreceptors and Eyes. *Evolutionary Biology*. , 207–263. (doi:10.1007/978-1-4615-6953-4_4)
11. Land MF, Fernald RD. 1992 The evolution of eyes. *Annu. Rev. Neurosci.* **15**, 1–29. (doi:10.1146/annurev.ne.15.030192.000245)
12. Suzuki TK. 2017 On the Origin of Complex Adaptive Traits: Progress Since the Darwin Versus Mivart Debate. *Journal of Experimental Zoology Part B: Molecular and Developmental Evolution*. **328**, 304–320. (doi:10.1002/jez.b.22740)
13. Nödl M-T, Fossati SM, Domingues P, Sánchez FJ, Zullo L. 2015 The making of an octopus arm. *EvoDevo*. **6**. (doi:10.1186/s13227-015-0012-8)
14. Nödl M-T, Kerbl A, Walzl MG, Müller GB, de Couet HG. 2016 The cephalopod arm crown: appendage formation and differentiation in the Hawaiian bobtail squid *Euprymna scolopes*. *Frontiers in Zoology*. **13**. (doi:10.1186/s12983-016-0175-8)
15. Tarazona OA, Lopez DH, Slota LA, Cohn MJ. 2019 Evolution of limb development in cephalopod mollusks. *Elife* **8**. (doi:10.7554/eLife.43828)
16. Wild E, Wollesen T, Haszprunar G, Heß M. 2015 Comparative 3D microanatomy and histology of the eyes and central nervous systems in coleoid cephalopod hatchlings. *Org. Divers. Evol.* **15**, 37–64. (doi:10.1007/s13127-014-0184-4)
17. 2001 Early Ontogeny of the Japanese Common Squid *Todarodes pacificus* (Cephalopoda, Ommastrephidae) with Special Reference to its Characteristic Morphology and Ecological Significance. *Zoolog. Sci.* **18**, 1011–1026. (doi:10.2108/zsj.18.1011)
18. Yamamoto M, Shimazaki Y, Shigeno S. 2003 Atlas of the embryonic brain in the pygmy squid, *Idiosepius paradoxus*. *Zoolog. Sci.* **20**, 163–179. (doi:10.2108/zsj.20.163)
19. Huminiecki L, Wolfe KH. 2004 Divergence of spatial gene expression profiles following species-specific gene duplications in human and mouse. *Genome Res.* **14**, 1870–1879.

(doi:10.1101/gr.2705204)

20. Liu S-L, Baute GJ, Adams KL. 2011 Organ and cell type-specific complementary expression patterns and regulatory neofunctionalization between duplicated genes in *Arabidopsis thaliana*. *Genome Biol. Evol.* **3**, 1419–1436. (doi:10.1093/gbe/evr114)
21. Li W-H, Yang J, Gu X. 2005 Expression divergence between duplicate genes. *Trends in Genetics.* **21**, 602–607. (doi:10.1016/j.tig.2005.08.006)
22. Lemer S, Bieler R, Giribet G. 2019 Resolving the relationships of clams and cockles: dense transcriptome sampling drastically improves the bivalve tree of life. *Proc. Biol. Sci.* **286**, 20182684. (doi:10.1098/rspb.2018.2684)
23. Buresi A, Andouche A, Navet S, Bassaglia Y, Bonnaud-Ponticelli L, Baratte S. 2016 Nervous system development in cephalopods: How egg yolk-richness modifies the topology of the mediolateral patterning system. *Dev. Biol.* **415**, 143–156. (doi:10.1016/j.ydbio.2016.04.027)